# ShiftAddViT: Mixture of Multiplication Primitives Towards Efficient Vision Transformers

**Haoran You**[*], **Huihong Shi**[*], **Yipin Guo**[*], and **Yingyan (Celine) Lin**

Georgia Institute of Technology, Atlanta, GA
*{haoran.you, celine.lin}@gatech.edu, eic-lab@groups.gatech.edu*

## Abstract

Vision Transformers (ViTs) have shown impressive performance and have become a unified backbone for multiple vision tasks. However, both the attention mechanism and multi-layer perceptrons (MLPs) in ViTs are not sufficiently efficient due to dense multiplications, leading to costly training and inference. To this end, we propose to reparameterize pre-trained ViTs with a mixture of multiplication primitives, e.g., bitwise shifts and additions, towards a new type of multiplication-reduced model, dubbed **ShiftAddViT**, which aims to achieve end-to-end inference speedups on GPUs without requiring training from scratch. Specifically, all `MatMuls` among queries, keys, and values are reparameterized using additive kernels, after mapping queries and keys to binary codes in Hamming space. The remaining MLPs or linear layers are then reparameterized with shift kernels. We utilize TVM to implement and optimize those customized kernels for practical hardware deployment on GPUs. We find that such a reparameterization on (quadratic or linear) attention maintains model accuracy, while inevitably leading to accuracy drops when being applied to MLPs. To marry the best of both worlds, we further propose a new mixture of experts (MoE) framework to reparameterize MLPs by taking multiplication or its primitives as experts, e.g., multiplication and shift, and designing a new latency-aware load-balancing loss. Such a loss helps to train a generic router for assigning a dynamic amount of input tokens to different experts according to their latency. In principle, the faster the experts run, the more input tokens they are assigned. Extensive experiments on various 2D/3D Transformer-based vision tasks consistently validate the effectiveness of our proposed ShiftAddViT, achieving up to **5.18×** latency reductions on GPUs and **42.9%** energy savings, while maintaining a comparable accuracy as original or efficient ViTs. Codes and models are available at `https://github.com/GATECH-EIC/ShiftAddViT`.

## 1 Introduction

Vision Transformers (ViTs) have emerged as powerful vision backbone substitutes to convolutional neural networks (CNNs) due to their impressive performance [57, 16]. However, ViTs' state-of-the-art (SOTA) accuracy comes at the price of prohibitive hardware latency and energy consumption for both training and inference, limiting their prevailing deployment and applications [34, 51], even though pre-trained ViT models are publicly available. The bottlenecks are twofold based on many profiling results: attention and MLPs, e.g., they account for 36%/63% of the FLOPs and 45%/46% of the latency when running DeiT-Base [55] on an NVIDIA Jetson TX2 GPU [72, 19]). Previous efficient ViT solutions mostly focus on macro-architecture designs [21, 39, 32, 55, 9, 61] and linear attention optimization [59, 50, 5, 13, 6, 73]. However, they pay less attention on reducing the underlying dominant multiplications as well as on co-optimizing both attention and MLPs.

We identify that one of the most effective yet still missing opportunities for accelerating ViTs is to reparameterize their redundant multiplications with a mixture of multiplication primitives, i.e.,

---

[*]Equal Contribution.

37th Conference on Neural Information Processing Systems (NeurIPS 2023).

bitwise shifts and adds. The idea is drawn from the common hardware design practice in computer architecture or digital signal processing. That is, the multiplication can be replaced by bitwise shifts and adds [67, 22]. Such a hardware-inspired approach can lead to an efficient and fast hardware implementation without compromising the model accuracy. As such, this paper embodies the shift&add idea in ViTs towards a new type of multiplication-reduced model. Moreover, unlike previous ShiftAddNet [70] which requires full and slow training and dedicated hardware accelerator support, this work starts from pre-trained ViTs to avoid the tremendous cost of training from scratch and targets end-to-end inference speedups, i.e., accelerating both attention and MLPs, on GPUs.

The aforementioned shift&add concept uniquely inspires our design of naturally hardware-friendly ViTs, but it also presents us with three challenges to address. *First*, how to effectively reparameterize ViTs with shifts and adds? Previous ShiftAddNet [70] reparameterizes CNNs by cascading shift layers and add layers, leading to a doubled number of layers or parameters. The customized CUDA kernels of shift and add layers also suffer from much slower training and inference than PyTorch [42] on GPUs. Both motivate a new easy-to-accelerate reparameterization method for ViTs. *Second*, how to maintain the accuracy after reparameterization? It is expected to see accuracy drops when multiplica-

Table 1: Hardware cost under 45nm CMOS.

| OPs | Format | Energy (pJ) | Area ($\mu$m$^2$) |
|---|---|---|---|
| **Mult.** | FP32 | 3.7 | 7700 |
| | FP16 | 0.9 | 1640 |
| | INT32 | 3.1 | 3495 |
| | INT8 | 0.2 | 282 |
| **Add** | FP32 | 1.1 | 4184 |
| | FP16 | 0.4 | 1360 |
| | INT32 | 0.1 | 137 |
| | INT8 | 0.03 | 36 |
| **Shift** | INT32 | 0.13 | 157 |
| | INT16 | 0.057 | 73 |
| | INT8 | 0.024 | 34 |

tions are turned into shift and add primitives [8, 11]. Most works compensate for the accuracy drops by enlarging model sizes leveraging the much improved energy efficiency advantages of shifts and adds [70], e.g., up to $196\times$ unit energy savings than multiplications as shown in Tab. 1. While in ViTs, where input images are split into non-overlapping tokens, one can uniquely leverage the inherent adaptive sensitivity among input tokens. In principle, the essential tokens containing target objects are expected to be processed using more powerful multiplications. In contrast, tokens with less important background information can be handled by cheaper primitives. Such a principle aligns with the spirit of recent token merging [3] and input adaptive methods [45, 68, 74] for ViTs. *Third*, how can we balance the loading and processing time for the aforementioned important and less-important input tokens? For ViTs, using a mixture of multiplication primitives can lead to varying processing speeds for sensitive and non-sensitive tokens. This needs to be balanced; otherwise, intermediate activations may take longer to synchronize before progressing to the next layer.

To the best of our knowledge, *this is the first attempt* to address the above three challenges and to incorporate the shift&add concept from the hardware domain in designing multiplication-reduced ViTs using a mixture of multiplication primitives. Specifically, we make the following contributions:

- We take inspiration from hardware practice to reparameterize pre-trained ViTs with a mixture of complementary multiplication primitives, i.e., bitwise shifts and adds, to deliver a new type of multiplication-reduced network, dubbed **ShiftAddViT**. All `MatMuls` in attention are reparameterized by additive kernels, and the remaining linear layers and MLPs are reparameterized by shift kernels. The kernels are built in TVM for practical deployments.

- We introduce a new mixture of experts (MoE) framework for ShiftAddViT to preserve accuracy post-reparameterization. Each expert represents either a multiplication or its primitives, such as shifts. Depending on the importance of a given input token, the appropriate expert is activated, for instance, multiplications for vital tokens and shifts for less crucial ones.

- We introduce a latency-aware load-balancing loss term within our MoE framework to dynamically allocate input tokens to each expert. This ensures that the number of tokens assigned aligns with the processing speeds of the experts, significantly reducing synchronization time.

- We conduct extensive experiments to validate the effectiveness of our proposed ShiftAddViT. Results on various 2D/3D Transformer-based vision models consistently show its superior efficiency, achieving up to **5.18**$\times$ latency reductions on GPUs and **42.9%** energy savings, while maintaining a comparable or even higher accuracy than the original ViTs.

## 2 Related Works

**Vision Transformers (ViTs).** ViTs [16, 55] split images into non-overlapping patches or tokens and have outperformed CNNs on a range of vision tasks. They use a simple encoder-only architecture, primarily made up of attention modules and MLPs. The initial ViTs necessitated costly pre-training

on expansive datasets [52], and subsequent advancements like DeiT [55] and T2T-ViT [75] alleviated this by refining training methodologies or crafting innovative tokenization strategies. Later on, a surge of new ViT architecture designs, such as PVT [60], CrossViT [7], PiT [26], MViT [18], and Swin-Transformer [35], has emerged to improve the accuracy-efficiency trade-offs using a pyramid-like architecture. Another emergent trend revolves around dynamic ViTs like DynamicViT [45], A-ViT [68], ToME [3], and MIA-Former [74], aiming to adaptively eliminate non-essential tokens. Our MoE framework resonates with this concept but addresses those tokens using low-cost multiplication primitives that run concurrently. For edge deployment, solutions such as LeViT [21], MobileViT [39], and EfficientViT [5, 50] employ efficient attention mechanisms or integrate CNN feature extractors. This approach propels ViTs to achieve speeds comparable to CNNs, e.g., MobileNets [47]. In contrast, our proposed ShiftAddViT is the first to draw from shift&add hardware shortcuts for reparameterizing pre-trained ViTs, boosting end-to-end inference speeds and energy efficiency without hurting accuracy. Therefore, ShiftAddViT is orthogonal to existing ViT designs and can be applied on top of them.

Tremendous efforts have been dedicated to designing efficient ViTs. For instance, to address the costly self-attention module, which possesses quadratic complexity in relation to the number of input tokens, various linear attentions have been introduced. These can be broadly divided into two categories: local attention [35, 2, 56] and kernel-based linear attention [30, 13, 65, 38, 5, 34, 1]. For the former category, Swin [35] computes similarity measures among neighboring tokens rather than all tokens. QnA [2] disseminates the attention queries across all tokens. MaxViT [56] employs block attention and integrates dilated global attention to capture both local and global information. Regarding the latter category, most methods approximate the softmax function [13, 30, 4] or the complete self-attention matrix [65, 38] using orthogonal features or kernel embeddings, allowing the computational order to switch from $(\mathbf{QK})\mathbf{V}$ to $\mathbf{Q}(\mathbf{KV})$. Furthermore, a few efforts target reducing the multiplication count in ViTs. For instance, Ecoformer [34] introduces a novel binarization paradigm through kernelized hashing for $\mathbf{Q}/\mathbf{K}$, ensuring that `MatMuls` in attentions become solely accumulations. Adder Attention [51] examines the challenges of integrating adder operations into attentions and suggests including an auxiliary identity mapping. ShiftViT [58] incorporates spatial shift operations into attention modules. Different from previous works, our ShiftAddViT focuses on not only attention but also MLPs towards end-to-end ViT speedups. All `MatMuls` can be transformed into additions/accumulations using quantization, eliminating the need for complex kernelized hashing (KSH) as in [34]. For the remaining linear layers in the attention modules or MLPs, they are converted to bitwise shifts or MoEs, rather than employing spatial shifts like [58] for attentions.

**Multiplication-less NNs.** Numerous efforts aim to minimize the dense multiplications, which are primary contributors to the time and energy consumption in CNNs and ViTs. In the realm of CNNs, binary networks [14, 29] binarize both activations and weights; AdderNet [8, 66, 62] fully replaces multiplications with additions with a commendable accuracy drop; Shift-based networks use either spatial shifts [64] or bitwise shifts [17] to replace multiplications. Recently, techniques initially developed for CNNs have found applications in ViTs. For instance, BiTs [36, 25] introduce binary transformers that maintain commendable accuracy; Shu et al. and Wang et al. also migrate the add or shift idea to ViTs with the scope limited to attentions [51, 58]. Apart from network designs, there are also neural architecture search efforts made towards both accurate and efficient networks [71, 31]. As compared to the most related work, i.e., ShiftAddNet [70] and ShiftAddNAS [71], our ShiftAddViT *for the first time* enables shift&add idea in ViTs without compromising the model accuracy, featuring three more characteristics that make it more suitable for practical applications and deployment: (1) starting from pre-trained ViTs to avoid the costly and tedious training from scratch; (2) focusing on optimization for GPUs rather than on dedicated hardware such as FPGAs/ASICs; (3) seamlessly compatible with the MoE framework to enable switching between the mixture of multiplication primitives, such token-level routing and parallelism are uniquely applicable and designed for ViTs.

## 3 Preliminaries

**Self-attention and Vision Transformers.** Self-attention is a core ingredient of Transformers [57, 16], and usually contains multiple heads H with each measuring pairwise correlations among all input tokens to capture global-context information. It can be defined as below:

$$\mathtt{Attn}(\mathbf{X}) = \mathtt{Concat}(\mathrm{H}_1, \cdots, \mathrm{H}_h)\mathbf{W}_O, \text{ where } \mathrm{H}_i = \mathtt{Softmax}\left(\frac{f_Q(\mathbf{X}) \cdot f_K(\mathbf{X})^T}{\sqrt{d_k}}\right) \cdot f_V(\mathbf{X}), \quad (1)$$

where $h$ denotes the number of heads. Within each head, input tokens $\mathbf{X} \in \mathbb{R}^{n \times d}$ of length $n$ and dimension $d$ will be linearly projected to query, key, and value matrices, i.e., $\mathbf{Q}, \mathbf{K}, \mathbf{V} \in \mathbb{R}^{n \times d_k}$,

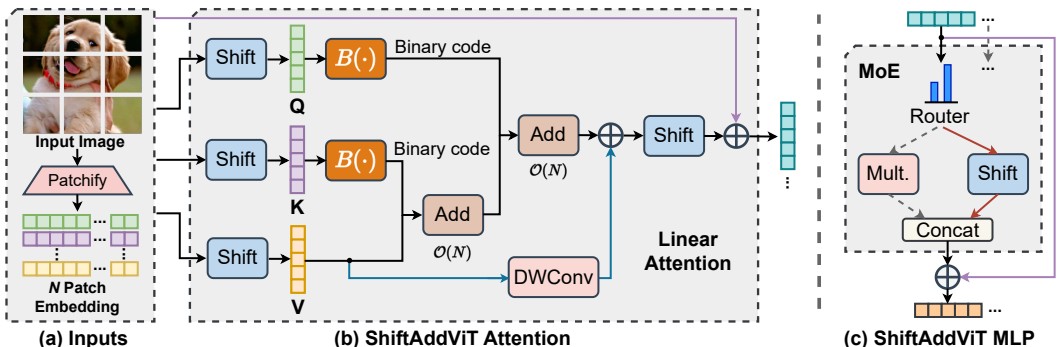

Figure 1: An illustration of the network architecture overview of ShiftAddViT.

through three linear mapping functions, $f_Q = \mathbf{X}\mathbf{W}_Q, f_K = \mathbf{X}\mathbf{W}_K, f_V = \mathbf{X}\mathbf{W}_V$, where $d_k = d/h$ is the embedding dimension of each head. The results from all heads are concatenated and projected with a weight matrix $\mathbf{W}_O \in \mathbb{R}^{d \times d}$ to generate the final outputs. Such an attention module is followed by MLPs with residuals to construct one transformer block that can be formulated as below:

$$\mathbf{X}_{\text{Attn}} = \texttt{Attn}(\texttt{LayerNorm}(\mathbf{X})) + \mathbf{X}, \quad \mathbf{X}_{\text{MLP}} = \texttt{MLP}(\texttt{LayerNorm}(\mathbf{X}_{\text{Attn}})) + \mathbf{X}_{\text{Attn}}. \quad (2)$$

**Shift and Add Primitives.** Direct hardware implementation of multiplications is inefficient. Shift and add primitives serve as "shortcuts" for streamlined hardware designs. Shift operations, equivalent to multiplying by powers of two, offer significant savings. As shown in Tab. 1, such shifts can reduce energy use by up to $23.8\times$ and chip area by $22.3\times$ compared to multiplications [33, 70], given the INT32 data format and 45nm CMOS technology. Similarly, add operations, another efficient primitive, can achieve up to $31.0\times$ energy and $25.5\times$ area savings relative to multiplications. Both primitives have driven numerous network architecture innovations [8, 17, 70, 71].

## 4 The Proposed ShiftAddViT

**Overview.** Given the widespread availability of pre-trained ViT model weights, we aim to fine-tune and transform them into multiplication-reduced ShiftAddViTs in order to decrease runtime latency and enhance energy efficiency. To achieve this, we must reparameterize both the attentions and MLPs in ViTs to shift and add operations. Instead of using cascaded shift layers and add layers as in ShiftAddNet [70] or designing specific forward and backpropagation schemes like AdderNet [8, 51], we lean on the simple yet effective approach of ViTs' long-dependency modeling capability. We propose replacing the dominant MLPs/Linears/MatMuls with shifts and adds, leaving the attention intact. This approach permits ShiftAddViTs to be adapted from existing ViTs without the need for training from scratch. As illustrated in Fig. 1, for attentions, we transform four Linear layers and two MatMuls into shift and add layers, respectively. For MLPs, direct replacement with shift layers results in significant accuracy degradation. Hence, we consider designing a dedicated MoE framework incorporating a blend of multiplication primitives, such as multiplication and shift, to achieve both accurate and efficient final models. These steps transition pre-trained ViTs into ShiftAddViTs, achieving much reduced runtime latency while maintaining a comparable or even higher accuracy.

### 4.1 ShiftAddViT: Reparameterization Towards Multiplication-reduced Networks

This subsection describes how we reparameterize pre-trained ViTs with hardware-efficient shift and add operations, including a detailed implementation and the corresponding sensitivity analysis.

**Reparameterization of Attention.** There are two categories of layers in attention modules: MatMuls and Linears, that can be converted to Add and Shift layers, respectively. In order to reparameterize MatMuls to Add layers, we consider performing binary quantization on one operand during MatMuls, e.g., $\mathbf{Q}$ or $\mathbf{K}$ for the MatMul of $\mathbf{QK}$, so that the multiply-and-accumulate (MAC) operations between two matrices will be replaced with merely energy-efficient add operations [34]. Furthermore, to build ShiftAddViT on top of efficient linear attentions [59, 34, 38], we exchange the order of MatMuls from $(\mathbf{QK})\mathbf{V}$ to $\mathbf{Q}(\mathbf{KV})$ for achieving linear complexity w.r.t. the number of input tokens. In this way, the binary quantization will be applied to $\mathbf{K}$ and $\mathbf{Q}$ as illustrated in Fig. 1 (b), leaving the more sensitive $\mathbf{V}$ branch as high precision. This also allows us to insert a lightweight depthwise convolution (DWConv) to $\mathbf{V}$ branch in parallel to enhance local feature capturing capability with negligible

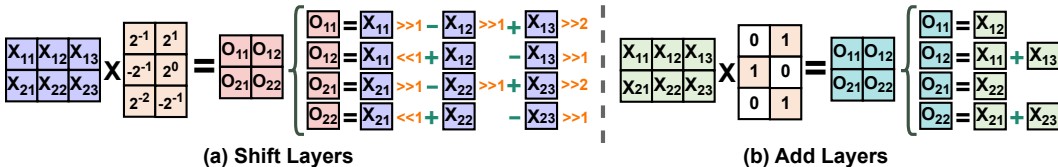

**(a) Shift Layers**    **(b) Add Layers**

Figure 2: Illustration of both `Shift` and `Add` layers, where $\mathbf{X}$ and $\mathbf{O}$ refer to inputs and outputs.

overhead ($<1\%$ of total MACs) following recent SOTA linear attention designs [65, 73]. On the other hand, for reparameterizing the left four Linears in the attention module with `Shift` layers, we resort to sign flips and power-of-two shift parameters to represent the Linear weights [17, 70].

We illustrate the implementation of `Shift` and `Add` layers in Fig. 2 and formulate them as follows:

$$\mathbf{O_{Shift}} = \sum \mathbf{X^T} \cdot \mathbf{W_S} = \sum \mathbf{X^T} \cdot \mathbf{s} \cdot 2^{\mathbf{P}}, \quad \mathbf{O_{Add}} = \sum \mathbf{X^T} \cdot \mathbb{1}\{\mathbf{W_A} \neq \mathbf{0}\}, \qquad (3)$$

where $\mathbf{X}$ refers to input activations, $\mathbf{W_S}$ and $\mathbf{W_A}$ are weights in `Shift` and `Add` layers, respectively. For `Shift` weights $\mathbf{W_S} = \mathbf{s} \cdot 2^{\mathbf{P}}$, $\mathbf{s} = sign(\mathbf{W}) \in \{-1, 1\}$ denotes sign flips, $\mathbf{P} = round(log_2(abs(\mathbf{W})))$, and $2^{\mathbf{P}}$ represents bitwise shift towards left ($\mathbf{P} > 0$) or right ($\mathbf{P} < 0$). Both $\mathbf{s}$ and $\mathbf{P}$ are trainable during the training. For `Add` weights, we have applied binary quantization and will skip zeros directly, the remaining weights correspond to additions or accumulations. Combining those two weak players, i.e., `Shift` and `Add`, leads to higher representation capability as also stated in ShiftAddNet [70], we highlight that our ShiftAddViT seamlessly achieves such combination for ViTs, without the necessity of cascading both of them as a substitute for one convolutional layer.

**Reparameterization of MLPs.** MLPs are often overlooked but also dominate ViTs' runtime latency apart from attentions, especially for the small or medium amount of tokens [19, 78, 43]. One natural thought is to reparameterize all MLPs with `Shift` layers as defined above. However, such aggressive design indispensably leads to severe accuracy drops, e.g., ↓1.18% for PVTv1-T [60], (DWConv used between two MLPs in PVTv2 are kept). We resolve this by proposing a new MoE framework to achieve speedups with comparable accuracy, as shown in Fig. 1 (c) and will be elaborated in Sec. 4.2.

**Sensitivity Analysis.** To investigate whether the above reparameterization methods work as expected, we break down all components and conduct sensitivity analysis. Each of them is applied to ViTs with only 30 epochs of finetuning. As summarized in Tab. 2, applying linear attention (LA), `Add`, or `Shift` to attention layers makes little influence on accuracy. But applying `Shift` to MLPs may lead to severe accuracy drops for sensitive models like PVTv1-T. Also, making MLPs more effi-

Table 2: Sensitivity analysis of reparameterizing ViTs with `Shift` and `Add`.

| Components | Apply | Acc. (%) of PVT [60, 61] | |
| --- | --- | --- | --- |
| | | **PVTv2-B0** | **PVTv1-T** |
| - | - | 70.50 | 75.10 |
| - | MSA | 71.25 | 76.21 |
| **Attention** | LA+Add | 70.95 | 75.20 |
| | Shift | 70.96 | 76.05 |
| **MLPs** | Shift | 70.28 | 73.92 |
| | MoE | 70.86 | 74.81 |

cient contribute a lot to the energy efficiency (e.g., occupy 65.7% on PVTv2-B0). Both the accuracy and efficiency perspectives motivate us to propose a new reparameterization scheme for MLPs.

## 4.2 ShiftAddViT: Mixture of Experts Framework

This subsection elaborates on the hypothesis and method of our ShiftAddViT MoE framework.

**Hypothesis.** We hypothesize there are important yet sensitive input tokens that necessitate the more powerful networks otherwise suffer from accuracy drops. Those tokens are likely to contain object information that directly correlates with our task objectives as validated by [3, 74, 45, 68]. While the left tokens are less sensitive and can be fully represented even using cheaper `Shift` layers. Such an input-adaptive nature of ViTs calls for a framework featuring a mixture of multiplication primitives.

**Mixture of Multiplication Primitives.** Motivated by the sensitivity analysis in Sec. 4.1, we consider two experts `Mult.` and `Shift` for processing important yet sensitive tokens and insensitive tokens, respectively. As illustrated in Fig. 1 (c), we apply the MoE framework to compensate for the accuracy drop. Each input token representation $\mathbf{x}$ will be passed to one expert according to the gate values $p_i = G(\mathbf{x})$ in routers, which are normalized via a softmax distribution $p_i := e^{p_i}/\sum_j^n e^{p_j}$ and will be jointly trained with model weights. The output can be formulated as $\mathbf{y} = \sum_i^n G(\mathbf{x}) \cdot E_i(\mathbf{x})$, where gate function $G(\mathbf{x}) = p_i(\mathbf{x}) \cdot \mathbb{1}\{p_i(\mathbf{x}) \geq p_j(\mathbf{x}), \forall j \neq i\}$, $n$ and $E_i$ denote the number of experts and $i$-th expert, respectively. Specifically, there are two obstacles when implementing such MoE in TVM:

- *Dynamic Input Allocation.* The input allocation is determined dynamically during runtime, and the trainable router within the MoE layers automatically learns this dispatch assignment as the gradients decay. We know the allocation only when the model is executed and we receive the router outputs. Therefore, the shape of expert input and corresponding indexes are dynamically changed. This can be handled by PyTorch with dynamic graphs while TVM expects static input shape. To this end, we follow and leverage the compiler support for dynamism as proposed in Nimble [49] on top of the Apache TVM to handle the dynamic input allocation.

- *Parallel Computing.* Different experts are expected to run in parallel, this can be supported by several customized distributed training frameworks integrated with PyTorch, e.g., FasterMoE [24], and DeepSpeed [44]. However, it remains nontrivial to support this using TVM. One option to simulate is to perform modularized optimization to mimic parallel computing, in particular, we optimize each expert separately and measure their latency, the maximum latency among all experts will be recorded and regarded as the latency of this MoE layer, and the aggregated total of the time spent for each layer is the final reported modularized model latency. To avoid any potential confusion between real-measured wall-clock time, i.e., no parallelism assumed, and simulated modularized latency, i.e., ideal parallelism assumed, we reported both for models containing MoE layers in Sec. 5 to offer more insights into the algorithm's feasibility and cost implications.

**Latency-aware Load-balancing Loss.** The key to our MoE framework is the design of a routing function to balance all experts towards higher accuracy and lower latency. Previous solutions [20, 23] use homogeneous experts and treat them equally. While in our MoE framework, the divergence and heterogeneity between powerful yet slow `Mult.` and fast yet less powerful `Shift` experts incur one unique question: *How to orchestrate the workload of each expert to reduce the synchronization time?* Our answer is a latency-aware load-balancing loss, which ensures (1) all experts receive the expected weighted sums of gate values; and (2) all experts are assigned the expected number of input tokens. Those two conditions are enforced by adopting importance and load balancing loss as defined below:

$$\mathcal{L}_{\texttt{IMP}} = \texttt{SCV}\left(\left\{\alpha_i \cdot \sum\nolimits_{\mathbf{x} \in \mathbf{X}} p_i(\mathbf{x})\right\}_{i=1}^n\right), \quad \mathcal{L}_{\texttt{LOAD}} = \texttt{SCV}\left(\left\{\alpha_i \cdot \sum\nolimits_{\mathbf{x} \in \mathbf{X}} q_i(\mathbf{x})\right\}_{i=1}^n\right), \quad (4)$$

where `SCV` denotes the squared coefficient of variation of given distributions over experts (also used in [46, 48]); $\alpha_i$ refers to the latency-aware coefficient of $i$-th expert and is defined by $^{(\texttt{Lat}_i)}/_{(\sum_j \texttt{Lat}_j)}$ because that the expected assignments are inversely proportional to runtime latency of $i$-th expert, $\texttt{Lat}_i$; $q_i(\mathbf{x})$ is the probability that the gate value of $i$-th expert outweighs the others, i.e., top1 expert, and is given by $\mathbf{P}\left(p_i(\mathbf{x}) + \epsilon \geq p_j(\mathbf{x}), \forall j \neq i\right)$. Note that this probability is discrete, we use a noise proxy $\epsilon$ following [48, 23] to make it differentiable. The above latency-aware importance loss and load-balanced loss help to balance gate values and workload assignments, respectively, and can be integrated with classification loss $\mathcal{L}_{\texttt{CLS}}$ as the total loss function $\mathcal{L}(\mathbf{X}) = \mathcal{L}_{\texttt{CLS}}(\mathbf{X}) + \lambda \cdot (\mathcal{L}_{\texttt{IMP}}(\mathbf{X}) + \mathcal{L}_{\texttt{LOAD}}(\mathbf{X}))$ for training ViTs and gates simultaneously. $\lambda$ is set as 0.01 for all experiments.

## 5 Experiments

### 5.1 Experiment Settings

**Models and Datasets.** *Tasks and Datasets.* We consider two representative 2D and 3D Transformer-based vision tasks to demonstrate the superiority of the proposed ShiftAddViT, including 2D image classification on ImageNet dataset [15] with 1.2 million training and 50K validation images and 3D novel view synthesis (NVS) task on Local Light Field Fusion (LLFF) dataset [40] with eight scenes. *Models.* For the classification task, we consider PVTv1 [60], PVTv2 [61], and DeiT [55]. For the NVS task, we consider Transformer-based GNT [53] with View- and Ray-Transformer models.

**Training and Inference Details.** *For the classification task,* we follow Ecoformer [34] to initialize the pre-trained ViTs with Multihead Self-Attention (MSA) weights, based on which we apply our reparameterization a two-stage finetuning: (1) convert MSA to linear attention [73] and reparameterize all `MatMuls` with add layers with 100 epoch finetuning, and (2) reparameterize MLPs or linear layers with shift or MoE layers after finetuning another 100 epoch. Note that we follow PVTv2 [61] and Ecoformer to keep the last stage as MSA for fair comparisons. *For the NVS task,* we still follow the two-stage finetuning but do not convert MSA weights to linear attention to maintain the accuracy. All experiments are run on a server with eight RTX A5000 GPUs with each having 24GB GPU memory.

**Baselines and Evaluation Metrics.** *Baselines.* For the classification task, we reparameterize and compare our ShiftAddViT with PVTv1 [60], PVTv2 [61], Ecoformer [34], and their MSA variants. For the NVS task, we reparameterize on top of GNT [53] and compare with both vanilla NeRF [41]

Table 4: Comparisons between ShiftAddViT and baselines. We show the breakdown analysis of ShiftAddViT with two kinds of $\mathbf{Q}/\mathbf{K}$ quantization. The throughput or latency is measured with a batch size of 32 or 1, where † denotes that the numbers are measured and reported after optimizing ShiftAddViT using TVM. We report both real-device and modularized latency for models with MoE.

| Methods | Linear Attn | Add | | Shift | MoE | PVTv2-B0 [61] | | | PVTv1-T [60] | | |
|---|---|---|---|---|---|---|---|---|---|---|---|
| | | KSH | Quant. | | | Acc. (%) | Lat. (ms) | T. (img./s) | Acc. (%) | Lat. (ms) | T. (img./s) |
| MSA | ✗ | ✗ | ✗ | ✗ | ✗ | 70.77 | 4.62 | 989 | 76.21 | 4.73 | 903 |
| PVT [61] | ✓ | ✗ | ✗ | ✗ | ✗ | 70.50 | 6.25 | 2227 | 75.10 | 5.78 | 1839 |
| PVT+MoE | ✓ | ✗ | ✗ | ✗ | ✓(MLPs) | 70.82 | 12.46 | 1171 | 75.27 | 10.91 | 834 |
| Ecoformer [34] | ✓ | ✓ | ✗ | ✗ | ✗ | 70.44 | 7.82 | 1348 | NaN | 7.43 | 1021 |
| | ✓ | ✓ | ✗ | ✗ | ✗ | 71.19 | 6.13 | 2066 | 75.50 | 5.78 | 1640 |
| | ✓ | ✓ | ✗ | ✗ | ✗ | 70.95 | 1.07† | 2530† | 75.20 | 1.42† | 1683† |
| ShiftAddViT | ✓ | ✓ | ✗ | ✓(Attn) | ✗ | 70.53 | 1.04† | 2447† | 74.77 | 1.39† | 1647† |
| (with KSH [34] | ✓ | ✓ | ✗ | ✓(Attn) | ✓(MLPs) | 70.16 | 1.39†/1.11* | N/A | 74.44 | 1.91†/1.21* | N/A |
| or Quant. [27] | ✓ | ✓ | ✗ | ✗ | ✓(Both) | 70.38 | 1.59†/1.20* | N/A | 74.73 | 2.12†/1.21* | N/A |
| to binarize Q/K) | ✓ | ✗ | ✗ | ✗ | ✗ | 71.36 | 6.34 | 2014 | 75.64 | 5.48 | 1714 |
| | ✓ | ✗ | ✓ | ✗ | ✗ | 71.04 | 1.00† | 2613† | 75.18 | 1.20† | 1907† |
| | ✓ | ✗ | ✓ | ✓(Both) | ✗ | 68.57 | 0.97† | 2736† | 73.47 | 1.18† | 1820† |
| | ✓ | ✗ | ✓ | ✗ | ✓(Both) | 70.59 | 1.51†/1.12* | N/A | 74.93 | 1.97†/1.02* | N/A |

* denotes the modularized latency simulated by separately optimizing each expert/router with ideal parallelism.

Table 5: Comparisons between ShiftAddViT and baselines. We apply our Shift&Add techniques on top of SOTA Transformer-based NeRF model, GNT [53], and report averaged PSNR (↑), SSIM (↑) and LPIPS (↓) on the LLFF dataset [40] across eight scenes. In addition, we also show the results on two representative scenes, Orchids, and Flower, and the rendering latency and energy measured on an Eyeriss-like accelerator. More results for other scenes are available in the Appendix C.

| Methods | Add | Shift | MoE | LLFF Averaged | | | Orchids | | | Flower | | | Lat. (s) | Energy (J) |
|---|---|---|---|---|---|---|---|---|---|---|---|---|---|---|
| | | | | PSNR | SSIM | LPIPS | PSNR | SSIM | LPIPS | PSNR | SSIM | LPIPS | | |
| NeRF [41] | - | - | - | 26.50 | 0.811 | 0.250 | 20.36 | 0.641 | 0.321 | 27.40 | 0.827 | 0.219 | 683.6 | 1065 |
| GNT [53] | - | - | - | 27.24 | 0.889 | 0.093 | 20.67 | 0.752 | 0.153 | 27.32 | 0.893 | 0.092 | 1071 | 1849 |
| ShiftAddViT | ✓ | ✗ | ✗ | 26.85 | 0.874 | 0.116 | 20.74 | 0.730 | 0.182 | 28.02 | 0.891 | 0.089 | 1108 | 1697 |
| | ✓ | ✓(Both) | ✗ | 26.85 | 0.875 | 0.116 | 20.78 | 0.730 | 0.182 | 28.05 | 0.892 | 0.088 | 568.5 | 844.0 |
| | ✓ | ✓(Attn) | ✓(MLPs) | 26.92 | 0.876 | 0.114 | 20.73 | 0.731 | 0.180 | 28.20 | 0.894 | 0.087 | 746.6 | 1093 |
| | ✗ | ✓(Both) | ✗ | 27.05 | 0.881 | 0.107 | 20.84 | 0.746 | 0.169 | 28.14 | 0.896 | 0.083 | 531.2 | 995.6 |

and GNT [53]. _Evaluation Metrics._ For the classification task, we evaluate the ShiftAddViT and baselines in terms of accuracy, GPU latency and throughput measured on an RTX 3090 GPU. For the NVS task, we evaluate all models in terms of PSNR, SSIM [63], and LPIPS [76]. We also measure and report the energy consumption of all the above models based on an Eyeriss-like hardware accelerator [12, 77], which calculates not only computational but also data movement energy.

## 5.2 ShiftAddViT over SOTA Baselines on 2D Tasks

To evaluate the effectiveness of our proposed techniques, we apply the ShiftAddViT idea to five commonly used ViT models, including DeiT [55] and various variants of PVTv1 [60] and PVTv2 [61]. We compare their performance with baselines on the image classification task. Tab. 3 highlights the overall comparison with the most competitive baseline, Ecoformer [34], from which we see that ShiftAddViT consistently outperforms all baselines in terms of accuracy-efficiency tradeoffs, achieving **1.74×** ∼ **5.18×** latency reduction on GPUs and **19.4%** ∼ **42.9%** energy savings measured

Table 3: Overall comparison between ShiftAddViT and the most competitive baseline on five models.

| Models | Methods | Acc. (%) | Latency (ms) | Energy (mJ) |
|---|---|---|---|---|
| PVTv2-B0 | Ecoformer [34] | 70.44 | 7.82 | 33.64 |
| | **ShiftAddViT** | **70.59** | **1.51** | **27.13** |
| PVTv1-T | Ecoformer [34] | NaN | 7.43 | 93.47 |
| | **ShiftAddViT** | **74.93** | **1.97** | **72.59** |
| PVTv2-B1 | Ecoformer [34] | 78.38 | 8.02 | 106.2 |
| | **ShiftAddViT** | **78.49** | **2.49** | **85.34** |
| PVTv2-B2 | Ecoformer [34] | 81.28 | 15.43 | 198.2 |
| | **ShiftAddViT** | **81.32** | **4.83** | **163.9** |
| DeiT-T | MSA [55] | 72.20 | 5.12 | 66.88 |
| | **ShiftAddViT** | **72.40** | **2.94** | **38.21** |

on the Eyeriss accelerator [12] with comparable or even better accuracy (↑**0.04%** ∼ ↑**0.20%**). Note that we apply MoE to both linear and MLP layers. Moreover, we further provide comparisons of a variety of ShiftAddViT variants and PVT with or without multi-head self-attention (MSA) and MoE in Tab. 4 and 6. We observe that our ShiftAddViT can be seamlessly integrated with linear attention [73, 65], simple or advanced $\mathbf{Q}/\mathbf{K}$ quantization [34, 27], and the proposed MoE framework, achieving up to **3.06×/4.14×/8.25×** and **2.47×/1.10×/2.09×** latency reductions and throughput improvements after the TVM optimization on an RTX 3090 as compared to MSA, PVT, and PVT+ MoE (note: two `Mult.` experts) baselines, respectively, under comparable accuracies, i.e., ±0.5%.

Table 6: Comparisons between ShiftAddViT and baselines on PVTv2-B1 and PVTv2-B2. The throughput or latency is measured with a batch size of 32 or 1 on an RTX 3090 GPU.

| Methods | Linear Attn | Add | | Shift | MoE | PVTv2-B1 [61] | | | PVTv2-B2 [61] | | |
|---|---|---|---|---|---|---|---|---|---|---|---|
| | | KSH | Quant. | | | Acc. (%) | Lat. (ms) | T. (img./s) | Acc. (%) | Lat. (ms) | T. (img./s) |
| MSA | ✗ | ✗ | ✗ | ✗ | ✗ | 78.83 | 4.77 | 821 | 81.82 | 9.08 | 508 |
| PVT [61] | ✓ | ✗ | ✗ | ✗ | ✗ | 78.70 | 6.75 | 1512 | 82.00 | 12.20 | 921 |
| Ecoformer [34] | ✓ | ✓ | ✗ | ✗ | ✗ | 78.38 | 8.02 | 874 | 81.28 | 15.43 | 483 |
| **ShiftAddViT** (with KSH [34] or Quant. [27] to binarize Q/K) | ✓ | ✗ | ✗ | ✗ | ✗ | 78.80 | 6.36 | 1373 | 81.40 | 9.21 | 796 |
| | ✓ | ✓ | ✗ | ✗ | ✗ | 78.70 | 1.70† | 923† | 81.31 | 3.18† | 518† |
| | ✓ | ✓ | ✗ | ✓(Attn) | ✓(MLPs) | 78.20 | 2.46†/1.46* | N/A | 81.06 | 4.17†/3.22* | N/A |
| | ✓ | ✓ | ✗ | ✗ | ✓(Both) | 78.38 | 2.64†/1.48* | N/A | 81.32 | 4.83†/3.35* | N/A |
| | ✓ | ✗ | ✗ | ✗ | ✗ | 78.93 | 6.49 | 1327 | 81.55 | 11.82 | 782 |
| | ✓ | ✗ | ✓ | ✗ | ✗ | 78.70 | 1.57† | 942† | 81.33 | 2.82† | 553† |
| | ✓ | ✗ | ✓ | ✓(Both) | ✗ | 77.55 | 1.54† | 1021† | 79.97 | 2.93† | 600† |
| | ✓ | ✗ | ✓ | ✗ | ✓(Both) | 78.49 | 2.49†/1.35* | N/A | 81.01 | 4.55†/3.16* | N/A |

**Training Wall-clock Time.** To offer more insights into the training cost, we collect the training wall-clock time, which is $21\% \sim 25\%$ less than training from scratch as the original Transformer model and more than $50\times$ less than the previous ShiftAddNet [70] did on GPUs. For example, training the PVTv1-Tiny model from scratch takes 62 hours while our finetuning only needs 46 hours.

**Linear Attention vs. MSA.** We ensure batch size (BS) is 1 for all latency measurements and find the counterintuitive phenomenon that PVT with linear attention is slower than PVT with MSA. The reasons are two folds: (1) linear attention introduces extra operations, e.g., normalization, DWConv, or spatial reduction block that cost more time under small BS; and (2) the linear complexity is w.r.t. the number of tokens while the input resolution of 224 is not sufficiently large, resulting in limited benefits. To validate both points, we measure the latency of PVTv2-B0 with various BS and input resolutions as shown in the Appendix F, the results indicate that linear attention's benefits show up under larger BS or input resolutions.

### 5.3 ShiftAddViT over SOTA Baselines on 3D Tasks

We also extend our methods to 3D NVS tasks and built ShiftAddViT on top of Transformer-based GNT models [53]. We test and compare the PSNR, SSIM, LPIPS, latency, and energy of both ShiftAddViT and baselines, i.e., vanilla NeRF [41] and GNT [53], on the LLFF dataset across eight scenes. As shown in Tab. 5, our ShiftAddViT consistently leads to better accuracy-efficiency tradeoffs, achieving **22.3%/50.4%** latency reductions and **20.8%/54.3%** energy savings under comparable or even better generation quality (↑0.55/↓0.19 averaged PSNR across eight scenes), as compared to NeRF and GNT baselines, respectively. Note that GNT costs more than NeRF because of an increased number of layers. In particular, ShiftAddViT even achieves better generation quality (up to ↑0.88 PSNR) than GNT on some representation scenes, e.g., Orchid and Flower, as highlighted in Tab. 5, where the first, second, and third ranking performance are noted with corresponding colors. The full results of eight scenes are supplied in the Appendix C. Note that we only finetune ShiftAddViT for 140K steps on top of pre-trained GNT models instead of training up to 840K steps from scratch.

### 5.4 Ablation Studies of ShiftAddViT

**Performance Breakdown Analysis.** To investigate how each of the proposed techniques contributes to the final performance, we conduct ablation studies among various kinds of ShiftAddViT variants on PVTv1-T, PVTv2-B0/B1/B2 models to gradually break down and show the advantage of each component. As shown in Tab. 4 and 6, we have three general observations: (1) ShiftAddViT is robust to the binarization method of $\mathbf{Q}/\mathbf{K}$, e.g., it is compatible with either KSH [34] or vanilla binarization [27], achieving on average **3.03×** latency reductions than original PVT under comparable accuracies ($\pm 0.5\%$), and both **3.29×/1.32×** latency/energy reductions and $0.04\% \sim 0.20\%$ accuracy improvements over Ecoformer [34]; (2) vanilla binarization methods work better than KSH in our ShiftAddViT framework in terms of both accuracy ($\downarrow 0.05\% \sim \uparrow 0.21\%$) and efficiency (on average ↓5.9% latency reductions and ↑5.8% throughput boosts). Moreover, KSH requires that $\mathbf{Q}$ and $\mathbf{K}$ are identical while vanilla binarization [27] does not have such a limitation. That explains why applying vanilla quantization to the linear attention layers results in an accuracy improvement of on average 0.15% as compared to using KSH. Also note that our adopting linear attention following [73] works better (↑0.29%) than PVT [60, 61]; (3) Replacing linear layers in attention modules or MLPs with Shift layers lead to an accuracy drop while our proposed MoE can help to compensate for it. Specifically, adopting Shift layers leads to on average 1.67% accuracy drop while MoE instead improves on average 1.37% accuracy. However, MoE hurts the efficiency of both the PVT baseline

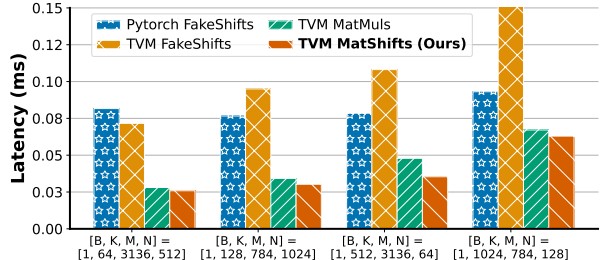

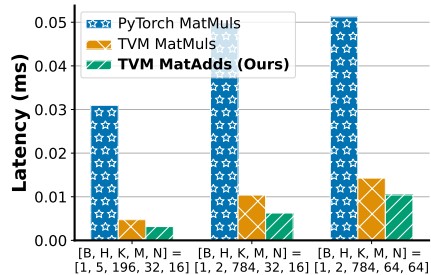

Figure 4: MLP/Linear latency speedups using shifts, where inputs are of shape $B \times K \times M$, weights are of shape $K \times N$. All dimensions are set w.r.t. the real dimensions in PVTs [60].

Figure 5: `MatMuls` latency speedups using adds, where inputs are of shape $B \times H \times K \times M$, weights are of shape $B \times H \times K \times N$.

and our ShiftAddViT without customized system acceleration, e.g., PVT+MoE increases 94% latency and reduces 51% throughput than PVT on GPUs, due to limited parallelism supports, especially for TVM. We report the modularized latency by separately optimizing each expert to demonstrate the potential of MoE's double-winning accuracy ($\uparrow 0.94\% \sim \uparrow 2.02\%$) and efficiency ($\downarrow 15.4\% \sim \uparrow 42.7\%$).

In addition, we want to clarify that the seemingly minor latency improvements of adopting shifts in Tab. 4 and 6 are due to full optimization of the compiled model as a whole (e.g., 6.34ms $\rightarrow$ 1ms for PVTv2-B0) on GPUs with sufficient chip area. Most gains are concealed by data movements and system-level schedules. Adopting shift layers substantially lowers energy and chip area usage as validated in Tab. 1 and the Appendix F. Under the same chip areas, latency savings of adopting shifts are more pertinent, e.g., PVTv2-B0 with shift or MoE achieve $3.9 \sim 5.7\times$ or $1.6 \sim 1.8\times$ speedups.

Apart from 2D tasks, we also report the performance breakdown on 3D NVS tasks, as shown in Tab. 5. The above three observations still hold except for the `Shift` layer, as we find that ShiftAddViT with all linear layers or MLPs replaced by `Shift` layers achieve even better PSNR ($\uparrow 0.13 \sim \uparrow 0.20$), SSIM ($\uparrow 0.005 \sim \uparrow 0.007$), and LPIPS ($\downarrow 0.007 \sim \downarrow 0.009$) than other variants. In addition, we profile on an Eyeriss

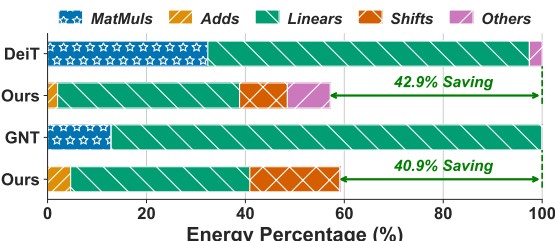

Figure 3: Energy breakdown on an Eyeriss accelerator.

accelerator to show the energy breakdown of both ShiftAddViT and baselines. As shown in Fig. 3, our ShiftAddViT reduced **42.9**% and **40.9**% energy on top of DeiT-T and GNT, respectively. Among them applying `Add` layers lead to **93.8**% and **63.8**% energy reductions than original `MatMuls`, `Shift` layers help to reduce **28.5**% and **37.5**% energy than previous linear/MLP layers. This set of experiments validates the effectiveness of each component in our proposed ShiftAddViT framework.

**Speedups of Shifts and Adds.** Apart from the overall comparison and energy reduction profiling, we also test the GPU speedups of our customized `Shift` and `Add` kernels, as shown in Fig. 4 and 5, respectively. We can see that both customized kernels achieve faster speeds than PyTorch and TVM baselines. For example, our `MatAdds` achieve on average **7.54**$\times$/**1.51**$\times$ speedups than PyTorch and TVM `MatMuls`, respectively. Our `MatShifts` achieve on average **2.35**$\times$/**3.07**$\times$/**1.16**$\times$ speedups than PyTorch `FakeShifts` [17], TVM `FakeShifts`, and TVM `MatMuls`, respectively. More comparisons and speedup analysis under larger batch sizes are supplied in the Appendix A. Note that in those comparisons, we use the default `einsum` operator for the PyTorch `MatMuls`, floating-point multiplication with power-of-twos for FakeShift, and implement the `MatAdds` and `MatShifts` using TVM [10] ourselves. We compare those customized operators with their multiplication counterparts in the same setting for fair comparisons. The speedup of `MatAdds` is because we replace multiplication-and-accumulation with accumulation only. The speedup of `MatShifts` is mainly attributed to the bit reductions (`INT32` and `INT8` for inputs and shift signs/weights) and thus data movement reductions instead of computations which are almost fully hidden behind data movements.

**MoE's Uniqueness in ShiftAddViT.** Our contributions are the multiplication-reduced ViTs and a new load-balanced MoE framework. They are linked in one organic whole. Unlike previous MoE works where all experts are of the same capacity and there is no distinction between tokens, our MoE layers combine unbalanced experts, i.e., multiplication and shift, with a hypothesis to divide important object tokens and background tokens. Such a new MoE design offers a softer version of

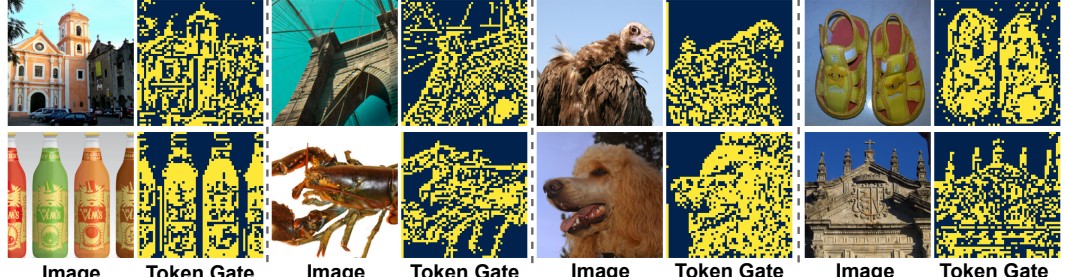

Figure 6: Visualization of the token dispatches in MoE routers/gates given images from the validation set, where yellow color denotes `Mult.` experts while blue color refers to `Shift` experts.

ShiftAddViTs, i.e., instead of aggressively replacing all multiplications with cheaper shifts and adds, we keep the powerful multiplication option to handle importance tokens for maintaining accuracy while leaving all remaining unimportant tokens being processed by cheaper bitwise shifts, winning a better accuracy and efficiency tradeoff.

**Ablation studies of Latency-aware Load-balancing Loss (LL-Loss).** To demonstrate the effectiveness of the proposed LL-Loss in our ShiftAddViT MoE framework, we conduct ablation studies on two PVT models as shown in Tab. 7. We find that ShiftAddViT w/ LL-Loss achieves better accuracy-efficiency tradeoffs, e.g., **14.6**% latency reductions while maintaining comparable accuracies ($\pm 0.1$%) when considering `Mult.` and `Shift`

Table 7: Ablation studies of the ShiftAd-dViT w/ or w/o LL-Loss on PVT models.

| Models | Methods | Acc. (%) | Norm. Latency |
|--------|---------|----------|---------------|
| PVTv2-B0 | w/o LL-Loss | 70.38 | 100% |
|          | **w/ LL-Loss** | **70.37** | **85.4%** |
| PVTv1-T | w/o LL-Loss | 74.73 | 100% |
|         | **w/ LL-Loss** | **74.66** | **85.5%** |

experts. The latency reduction could be larger if we consider more unbalanced experts with distinct runtimes. This set of experiments justifies the effectiveness of our LL-Loss in the MoE system.

**Visualization of Token Dispatches in MoE.** To validate our hypothesis that important yet sensitive tokens require powerful experts while other insensitive tokens can be represented with much cheaper `Shift` experts, we visualize the token dispatches in MoE routers/gates of the first `MLP` layer in our ShiftAddViT PVTv2-B0 model as shown in Fig. 6. We see that our designed router successfully identifies the important tokens that contain object information, which are then dispatched to more powerful `Multiplication` experts, leaving unimportant tokens, e.g., background, to cheaper `Shift` tokens. This set of visualization further validates the hypothesis and explains why our proposed MoE framework can effectively boost our ShiftAddViT towards better accuracy-efficiency trade-offs.

### 5.5 Limitation and Societal Impact Discussion

We made a firm step to show the practical usage of the proposed hardware-inspired ShiftAddViT. While this relies on dedicated TVM optimization, the full potential can be unraveled with customized hardware accelerators toward naturally hardware-efficient ViTs. Also, the unbalanced MoE framework shows its generalizability but highly demands system support with ideal parallelism.

## 6 Conclusion

In this paper, we for the first time propose a hardware-inspired multiplication-reduced ViT model dubbed ShiftAddViT. It reparameterizes both attention and MLP layers in pre-trained ViTs with a mixture of multiplication primitives, e.g., bitwise shifts and adds, towards end-to-end speedups on GPUs and dedicated hardware accelerators without the need for training from scratch. Moreover, a novel mixture of unbalanced experts framework equipped with a new latency-aware load-balancing loss is proposed in pursuit of double-winning accuracy and hardware efficiency. We use multiplication or its primitives as experts in our ShiftAddViT cases. Extensive experiments on both 2D and 3D Transformer-based vision tasks consistently validate the superiority of our proposed ShiftAddViT as compared to multiple ViT baselines. We believe that this paper opens up a new perspective on designing energy-efficient ViT inference based on widely available pre-trained ViT models.

## Acknowledgement

The work is supported in part by the National Science Foundation (NSF) through the RTML program (Award number: 1937592) and CoCoSys, one of the seven centers in JUMP 2.0, a Semiconductor Research Corporation (SRC) program sponsored by DARPA.

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

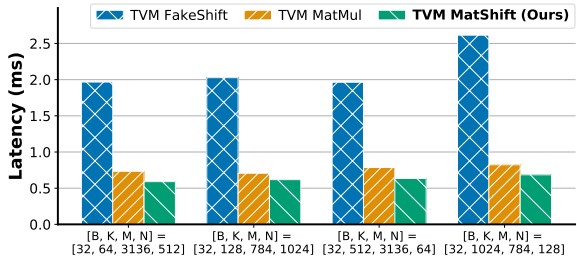
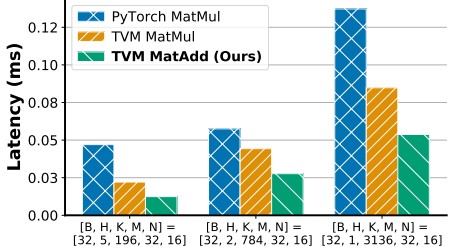

Figure 7: MLP/Linear latency speedups using shifts, where inputs are of shape $B \times K \times M$, weights are of shape $K \times N$. All dimensions are set w.r.t. the real dimensions in PVTs [60].

Figure 8: `MatMuls` latency speedups using adds, where inputs are of shape $B \times H \times K \times M$, weights are of shape $B \times H \times K \times N$.

## A  More Speedup Comparisons of `Shift` and `Add`

Apart from the speedup analysis when the batch size is one, we also conducted GPU speedup tests on our customized `Shift` and `Add` kernels when using larger batch sizes, such as 32, as depicted in Fig. 7 and 8, respectively. The results demonstrate that both our customized kernels consistently outperform the PyTorch and TVM baselines in terms of speed. For instance, our `MatAdds` achieve an average speedup of **2.80×**/**1.65×** compared to PyTorch and TVM `MatMuls`, respectively. Similarly, our `MatShifts` achieve an average speedup of **3.38×**/**1.21×** compared to TVM `FakeShifts` [17] and TVM `MatMuls`, respectively. It is important to note that in these comparisons, we ensured fair conditions by using the default `einsum` operator for PyTorch `MatMuls`, employing floating-point multiplication with power-of-twos for FakeShift, and implementing the `MatAdds` and `MatShifts` ourselves using TVM [10]. We compared these customized operators with their multiplication counterparts in the same setup to ensure a fair evaluation. The observed speedup of `MatAdds` is due to the replacement of multiplication-and-accumulation with accumulation only. On the other hand, the speedup of `MatShifts` is primarily attributed to the reduction in bit precision (`INT32` and `INT8` for inputs and shift signs/weights, respectively), resulting in reduced data movement instead of computations. These computational improvements are almost entirely hidden behind data movements.

## B  More Visualization of ShiftAddViT MoE Framework

In order to validate our hypothesis that important yet sensitive tokens require powerful experts, and other insensitive tokens can be represented by cheaper `Shift` experts, we have conducted additional token dispatch visualizations in the MoE routers/gates of the first `MLP` layer in our ShiftAddViT PVTv2-B0 model, as illustrated in Fig. 9. These visualizations provide further evidence that our designed routers successfully identify the crucial tokens that contain object information. Consequently, these tokens are dispatched to more powerful `Mult.` experts, while less important tokens, such as background tokens, are assigned to cheaper `Shift` experts. This collection of visualizations further reinforces our hypothesis and elucidates why our proposed MoE framework effectively enhances the accuracy-efficiency trade-offs of our ShiftAddViT model.

## C  More Results on 3D Tasks

We have extended our methods to 3D NVS tasks and have built ShiftAddViT on top of Transformer-based GNT models [53]. In the main paper, we tested and reported the averaged PSNR, SSIM, LPIPS, as well as latency and energy of both ShiftAddViT and baselines, i.e., the vanilla NeRF [41] and GNT [53], on the LLFF dataset across eight scenes. Here, we further supply the full results of eight scenes. As shown in Tab. 8, 9, 10, our ShiftAddViT consistently leads to better accuracy-efficiency tradeoffs in terms of the three metrics mentioned above, achieving comparable or even better generation quality (↑0.55/↓0.19 averaged PSNR across eight scenes), as compared to NeRF and GNT baselines, respectively. In particular, ShiftAddViT achieves better generation quality (up to ↑0.88 PSNR) than GNT on some representation scenes, e.g., Orchid and Flower. In all tables, the first , second , and third ranking performances are noted with corresponding colors.

In addition, we supply more qualitative rendering visualization on three scenes, Horns, Orchids, and Flowers, as shown in Fig. 10. We see that our ShiftAddViT generates more clear images than vanilla

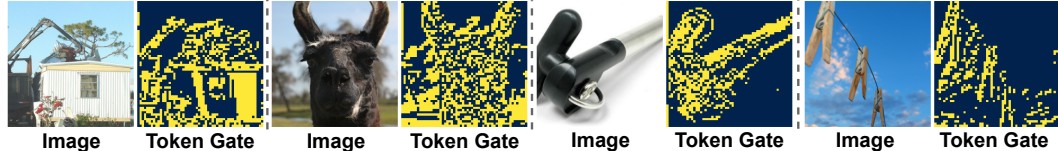

**Figure 9:** Visualization of the token dispatches in MoE routers/gates given images from the validation set, where yellow color denotes `Multiplication` experts and blue color refers to `Shift` experts.

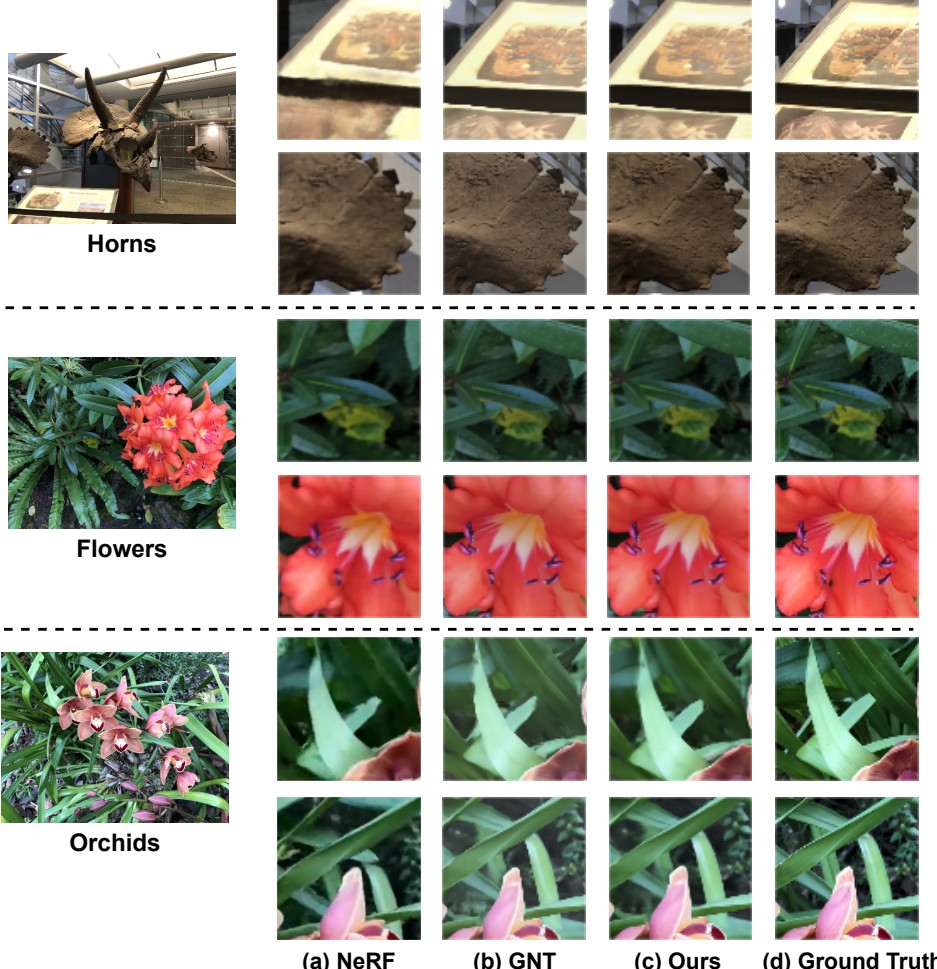

**Figure 10:** Qualitative comparisons between (a) NeRF [41], (b) GNT [53], (c) ShiftAddViT (Ours), and (d) ground truth when tested on the LLFF dataset for single-scene rendering.

NeRF [41] and comparable or even better details than GNT [53]. Quantitative and qualitative results consistently show that our ShiftAddViT does not hurt rendering quality when improving rendering speeds and energy efficiency.

## D   More Results on NLP Tasks

We also test our proposed optimization of MHSA and MLPs to NLP tasks. In particular, we apply our methods to Transformer models and Long Range Arena (LRA) benchmarks consisting of sequences ranging from 1K to 16K tokens [54]. The results are shown in Tab. 11. We see that our proposed ShiftAdd Transformers consistently show superior performance in terms of both model accuracy (+0.45% ∼ +5.79%) and efficiency (1.5× ∼ 11.5× latency reductions and 2.2× ∼ 16.4× energy reductions on an Eyeriss-like accelerator) as compared to original Transformer and other linear

Table 8: Comparisons between ShiftAddViT and baselines. We apply our Shift&Add techniques on top of the SOTA Transformer-based NeRF model, GNT [53], and report averaged PSNR (↑) on the LLFF dataset [40] across eight scenes.

| Methods | Add | Shift | MoE | PSNR (↑) on LLFF Dataset | | | | | | | |
| --- | --- | --- | --- | --- | --- | --- | --- | --- | --- | --- | --- |
| | | | | Room | Fern | Leaves | Fortress | Orchids | Flower | T-Rex | Horns |
| NeRF [41] | - | - | - | 32.70 | 25.17 | 20.92 | 31.16 | 20.36 | 27.40 | 26.80 | 27.45 |
| GNT [53] | - | - | - | 32.96 | 24.31 | 22.57 | 32.28 | 20.67 | 27.32 | 28.15 | 29.62 |
| ShiftAddViT | ✓ | ✗ | ✗ | 31.68 | 24.63 | 22.34 | 31.79 | 20.74 | 28.02 | 26.99 | 28.62 |
| | ✓ | ✓(Both) | ✗ | 31.64 | 24.63 | 22.34 | 31.73 | 20.78 | 28.05 | 27.03 | 28.61 |
| | ✓ | ✓(Attn) | ✓(MLPs) | 31.85 | 24.65 | 22.37 | 31.85 | 20.73 | 28.20 | 27.08 | 28.62 |
| | ✗ | ✓(Both) | ✗ | 32.12 | 24.35 | 22.42 | 32.15 | 20.84 | 28.14 | 27.32 | 29.09 |

Table 9: Comparisons between ShiftAddViT and baselines. We apply our Shift&Add techniques on top of the SOTA Transformer-based NeRF model, GNT [53], and report averaged SSIM (↑) on the LLFF dataset [40] across eight scenes.

| Methods | Add | Shift | MoE | SSIM (↑) on LLFF Dataset | | | | | | | |
| --- | --- | --- | --- | --- | --- | --- | --- | --- | --- | --- | --- |
| | | | | Room | Fern | Leaves | Fortress | Orchids | Flower | T-Rex | Horns |
| NeRF [41] | - | - | - | 0.948 | 0.792 | 0.690 | 0.881 | 0.641 | 0.827 | 0.880 | 0.828 |
| GNT [53] | - | - | - | 0.963 | 0.846 | 0.852 | 0.934 | 0.752 | 0.893 | 0.936 | 0.935 |
| ShiftAddViT | ✓ | ✗ | ✗ | 0.949 | 0.839 | 0.838 | 0.922 | 0.730 | 0.891 | 0.916 | 0.911 |
| | ✓ | ✓(Both) | ✗ | 0.948 | 0.839 | 0.839 | 0.923 | 0.730 | 0.892 | 0.917 | 0.911 |
| | ✓ | ✓(Attn) | ✓(MLPs) | 0.950 | 0.840 | 0.841 | 0.923 | 0.731 | 0.894 | 0.917 | 0.912 |
| | ✗ | ✓(Both) | ✗ | 0.955 | 0.836 | 0.841 | 0.930 | 0.746 | 0.896 | 0.923 | 0.922 |

Table 10: Comparisons between ShiftAddViT and baselines. We apply our Shift&Add techniques on top of the SOTA Transformer-based NeRF model, GNT [53], and report averaged LPIPS (↓) on the LLFF dataset [40] across eight scenes.

| Methods | Add | Shift | MoE | LPIPS (↓) on LLFF Dataset | | | | | | | |
| --- | --- | --- | --- | --- | --- | --- | --- | --- | --- | --- | --- |
| | | | | Room | Fern | Leaves | Fortress | Orchids | Flower | T-Rex | Horns |
| NeRF [41] | - | - | - | 0.178 | 0.280 | 0.316 | 0.171 | 0.321 | 0.219 | 0.249 | 0.268 |
| GNT [53] | - | - | - | 0.060 | 0.116 | 0.109 | 0.061 | 0.153 | 0.092 | 0.080 | 0.076 |
| ShiftAddViT | ✓ | ✗ | ✗ | 0.087 | 0.141 | 0.129 | 0.080 | 0.182 | 0.089 | 0.107 | 0.110 |
| | ✓ | ✓(Both) | ✗ | 0.088 | 0.142 | 0.128 | 0.080 | 0.182 | 0.088 | 0.106 | 0.110 |
| | ✓ | ✓(Attn) | ✓(MLPs) | 0.086 | 0.140 | 0.127 | 0.079 | 0.180 | 0.087 | 0.104 | 0.109 |
| | ✗ | ✓(Both) | ✗ | 0.076 | 0.138 | 0.125 | 0.070 | 0.169 | 0.083 | 0.097 | 0.097 |

attention baselines, which means that our shift and add reparameterization and load-balanced MoE ideas are generally applicable to Transformer models and are agnostic to domains and tasks.

# E   More Experiment Settings

*For the classification task,* we first replace the original attention layers in PVTv1 [60] and PVTv2 [61] with the standard MSA, and then finetune the MSA-based variants on top of the ImageNet-1k-pretrained weights for 100 epochs with a learning rate (lr) of $5 \times 10^{-5}$ following Ecoformer [34]. After that, we reparameterize the MSA-based models through a two-stage finetuning process: (1) convert MSA to linear attention [73] and reparameterize `MatMuls` with add layers via 100 epochs of finetuning with a base lr of $1 \times 10^{-5}$, and (2) reparameterize MLPs or linear layers with shift or MoE layers through another 100 epochs of finetuning with a base lr of $1 \times 10^{-5}$. All models are trained using eight RTX A5000 GPUs with a total batch size of 256, and we use the AdamW optimizer [37] with a cosine decay lr scheduler. All other hyperparameters are the same as those in Ecoformer [34] and PVTv2 [61]. *For the NVS task,* we apply reparameterization on top of the pretrained models from GNT [53] using a similar two-stage fine-tuning process, except that we do not convert MSA to linear attention for maintaining accuracy. Specifically, we first (1) reparameterize `MatMuls` with add layers and then (2) reparameterize MLPs or linear layers with shift or MoE layers. Both stages are finetuned for 140K steps with a base lr of $5 \times 10^{-4}$, and we sample 2048 rays with 192 coarse points sampled per ray in each iteration. All other hyperparameters are the same as those in GNT [53], including the use of the Adam optimizer with an exponential decay lr scheduler.

**Scaling Factor Clarification.** For shift layers, we reparameterize shifts following DeepShift-PS [17] and do not use a scaling factor. A straight-through estimator (STE) [69] is adopted to make the shift layer differentiable. For ShiftAddViT with KSH [34], there is no scaling factor needed as a set of

Table 11: Comparisons with our proposed ShiftAdd Transformer with baselines on Long Range Arena (LRA) benchmarks. Latency and energy are measured on an Eyeriss-like accelerator with a batch size of 1.

| Models | Accuracy (%) | | | | | Average Latency (ms) | Average Energy (mJ) |
|---|---|---|---|---|---|---|---|
| | Text (4K) | Listops (2K) | Retrieval (4K) | Image (1K) | Average | | |
| Transformer | 65.02 | 37.10 | 79.35 | 38.20 | 54.92 | 84.54 | 139.83 |
| Reformer | 64.88 | 19.05 | 78.64 | 43.29 | 51.47 | 11.19 | 19.04 |
| Linformer | 55.91 | 37.25 | 79.37 | 37.84 | 52.59 | 12.13 | 19.68 |
| Performer | 63.81 | 18.80 | 78.62 | 37.07 | 49.58 | 11.93 | 18.74 |
| **ShiftAdd-Transformer** | 66.69 | 37.15 | 82.02 | 35.62 | 55.37 | 7.38 | 8.53 |

Table 12: Latency comparisons of PVTv2-B0 with different attention types when measured under various batch sizes (BS) and input resolutions on an RTX 3090 GPU. "Linear SRA" donates the linear spatial-reduction attention in PVTv2.

| Attention Types | Pytorch Latency (ms) | | | | | | | | | | |
|---|---|---|---|---|---|---|---|---|---|---|---|
| | Input Resolution = (224, 224) | | | | | | | Input Resolution = (448, 448) | | | |
| | BS = 1 | BS = 2 | BS = 4 | BS = 8 | BS = 16 | BS = 32 | BS = 64 | BS = 1 | BS = 2 | BS = 4 | BS = 8 |
| MSA | 4.62 | 4.90 | 5.09 | 8.48 | 16.61 | 32.49 | 64.31 | 16.20 | 32.10 | 63.96 | 126.81 |
| PVTv2 (Linear SRA) | 6.25 | 6.33 | 6.73 | 6.75 | 6.79 | 11.68 | 22.22 | 6.36 | 6.40 | 6.84 | 12.96 |
| Linear Attention | 6.34 | 6.73 | 6.73 | 6.75 | 7.19 | 13.56 | 25.91 | 6.56 | 6.58 | 7.75 | 14.02 |

hash functions is applied to convert Q/K to binary codes. For ShiftAddViT with Quant. [27], we leverage layer-wise Quant. for both Q & K, the scaling factor can be multiplied after add operations. It can be efficiently implemented following [28].

# F   More Ablation Studies

**Linear Attention vs. MSA under Various Batch Sizes and Input Resolutions.** We find the counterintuitive phenomenon that PVT with linear attention is slower than PVT with MSA in the Tab. 4 and 6. To investigate the hidden reasons, we further measure the latency of PVTv2-B0 with various BS and input resolutions, the results are summarized in the Tab. 12. From that, we observe that linear attention's benefits show up under larger BS or input resolutions. The hidden reasons for this phenomenon are two folds: (1) linear attention introduces extra operations, e.g., normalization, DWConv, or spatial reduction block that cost more time under small BS; and (2) the linear complexity is w.r.t. the number of tokens while the input resolution of 224 is not sufficiently large, resulting in limited benefits.

**The Advantage of Using Shift.** Replacing Multiplication-based MLPs with shift layers significantly reduces latency, energy, and chip area costs. Our shift kernel offers an average speedup of $2.35\times/3.07\times/1.16\times$ compared to PyTorch FakeShift, TVM FakeShift, and TVM MatMuls, respectively, as shown in Fig. 3. The seemingly minor latency improvements in Tab. 4 and 6 are due to full optimization of the compiled model as a whole (e.g., 6.34ms → 1ms for PVTv2-B0) on GPUs with sufficient chip area. Most gains are concealed by data movements and system-level schedules. Adopting shift layers substantially lowers energy and chip area usage as validated in Tab. 1. Under the same chip areas, latency savings are more pertinent, e.g., PVTv2-B0 w/ shift or MoE achieve 3.9 $\sim 5.7\times$ and $1.6 \sim 1.8\times$ speedups, respectively, as summarized in Tab. 13.

Table 13: Comparisons of our ShiftAddViT with baselines in terms of task accuracy, latency measured with a batch size of 1 on an RTX 3090 GPU, and latency profiled on the Eyeriss accelerator under the same hardware area constraint.

| Model | Linear Attention + Add | Shift | MoE | Accuracy (%) | GPU Latency (ms) | Eyeriss Latency (ms) (Under the Same Chip Area) |
|---|---|---|---|---|---|---|
| PVTv2-B0 | ✗ | ✗ | ✗ | 70.77 | 4.62 (Pytorch) | 60.50 |
| **ShiftAddViT (PVTv2-B0)** | ✓ | ✗ | ✗ | 71.04 | 1.00 | 15.87 |
| | ✓ | ✓(Attn & MLP) | ✗ | 68.57 | 0.97 | 2.77 |
| | ✓ | ✗ | ✓(Attn & MLP) | 70.59 | 1.51 / 1.12* | 9.77 |
| PVTv2-B1 | ✗ | ✗ | ✗ | 78.83 | 4.77 (PyTorch) | 147.77 |
| **ShiftAddViT (PVTv2-B1)** | ✓ | ✗ | ✗ | 78.70 | 1.57 | 58.64 |
| | ✓ | ✓(Attn & MLP) | ✗ | 77.55 | 1.54 | 12.42 |
| | ✓ | ✗ | ✓(Attn & MLP) | 78.49 | 2.49 / 1.35* | 33.71 |

