# OpenReview forum: "ShiftAddViT: Mixture of Multiplication Primitives Towards Efficient Vision Transformer"
_NeurIPS.cc/2023/Conference — NeurIPS 2023 poster_

### Official Review · Reviewer_urMu · 2023-07-01

**Soundness:** 3 good
**Presentation:** 3 good
**Contribution:** 2 fair
**Rating:** 6
**Confidence:** 4

**Summary:**

This paper proposes an efficient accelerating framework for vision transformer, ShiftAddViT, which reparameterizes pre-trained ViTs with a mixture of complementary multiplication primitives and MoE designs. Specifically, All MatMuls in self-attention modules are reparameterized by additive kernels, and the remaining linear layers and MLPs are reparameterized by shift kernels. Besides, they develop a new MoE system for maintaining accuracy after reparameterization, and use a latency-aware load-balancing loss term to assign a dynamic amount of input tokens to each expert. Extensive experiments on various 2D/3D Transformer-based vision models demonstrate their superiority and efficiency.

**Strengths:**

This paper is well-written and easy to follow. Converting multiplication operations to additive and shift is a promising technique in model compression and accleration, this paper provides a systematic design solutions for vision transformer architectures, which is interesting.
Experiments in this paper are sufficient, including 2D and 3D task., and they provides detailed latency comparison for different methods, which much helps prove their effectiveness.

**Weaknesses:**

ShiftAddViT is a complex system design solution, where the acceleration comes from the replacing of additive and shift operations with multiplication, the maintaining accuracy relies on the MoE design. And ShiftAddViT utils TVM to implement and optimize their customized kernels for practical hardware deployment on GPUs. In this paper, the contributions of multiplation less and MoE are much independent, this paper  seems not much suitable for algorithm-based conferences like NeurIPS/CVPR/ICML.

**Questions:**

1. What is the most important contributions in this paper？Multiplacation-less solution and MoE seems indepedent, It is a common sence that MoE could boost the performance on various tasks, improving accuracy with MoE seems unnecessary in this paper.

2. Re-parameterizing usually refers to one kind of mathematical equivalence transformations, such as RepVGG [1], AC-Net [2], what is the re-parameterizing in this paper?

3. ShiftAddNet [3] provides the first scheme for the replacing of shift and additive opration with multiplication, so what is the weakness for comination of ShiftAddNet  and ViT architectures, could you show the comparision with ShiftAddViT.

4. When quantized to 4/8-bit, original ViT models could also achieve high performance and better acceleration, could you compare the quantized ViT with ShiftAddViT?

5. More complex task is better, such as detection and segementation.

[1] Ding X, Zhang X, Ma N, et al. Repvgg: Making vgg-style convnets great again[C]//Proceedings of the IEEE/CVF conference on computer vision and pattern recognition. 2021: 13733-13742.

[2] Ding X, Guo Y, Ding G, et al. Acnet: Strengthening the kernel skeletons for powerful cnn via asymmetric convolution blocks[C]//Proceedings of the IEEE/CVF international conference on computer vision. 2019: 1911-1920.

[3]You H, Chen X, Zhang Y, et al. Shiftaddnet: A hardware-inspired deep network[J]. Advances in Neural Information Processing Systems, 2020, 33: 2771-2783.

**Limitations:**

This paper provides the the limitation and societal impact discussion for their proposed technique.

---

> ### Author Rebuttal · Authors · 2023-08-09
>
> We greatly appreciate your careful review and constructive suggestions. Below are our detailed responses to your concerns.
>
> **W1: The contributions of multiplication less and MoE are independent? Not very suitable for algorithm conferences like NeurIPS/CVPR/ICML?**
>
> We thank the reviewer for acknowledging our system design and practical deployment on GPUs.
>
> *As for the contribution*, we want to clarify that multiplication-less and MoE are linked in one organic whole. Unlike previous MoE works where all experts are of the same capacity and there is no distinction between tokens, our MoE layers combine unbalanced experts, i.e., multiplication and shift, with a hypothesis to divide important object tokens and background tokens as also visualized in Figure 6. Such a new MoE design offers a softer version of ShiftAddViTs, i.e., instead of aggressively replacing all multiplications with cheaper shifts and adds, we keep the powerful multiplication option to handle importance tokens for maintaining accuracy while leaving all remaining unimportant tokens being processed by cheaper bitwise shifts, winning a better accuracy and efficiency tradeoff.
>
> *As for the venues*, we humbly clarify that the topic of multiplication-reduced network design is suitable and of great interest to algorithm conference audiences. For example, AdderNet [CVPR’20 Oral], DeepShift [CVPRW’21], ShiftAddNet [NeurIPS’20], ShiftAddNAS [ICML’22], and Ecoformers [NeurIPS’22] adopt shift or add layers to replace multiplication-based operations in CNNs or Transformers. TVM optimization for real hardware deployment is used by previous works like HAWQ-V3 [ICML’21] as well.
>
> ---
> **Q1: What are the most important contributions? Multiplication-less solution and MoE seem independent? MoE’s uniqueness in this work?**
>
> Our contributions are the multiplication-reduced ViTs and a new load-balanced MoE framework for a soft alternative to ShiftAddViTs. Also, we provide the first-time systematic investigation of layer sensitivity, accuracy impact, allocation strategy, and hardware implementations when considering shift&add-based ViTs as also acknowledged by Reviewer FYmk.
>
> The multiplication-less solution and MoE are effectively combined, and the load-balanced and input-adaptive MoE design is uniquely applicable to our ShfitAddViTs, please also refer to our reply to your W1 for a detailed analysis.
>
> ---
> **Q2: What is the re-parameterizing in this paper?**
>
> Sorry for not making it clear enough. We are not making our model arithmetically identical to the original ViTs. Instead, we inherit the pre-trained weight to parameterize shift or add layers. For example, we follow the below equation to reparameterize the shift weights based on the inherited weights:
>
> $W_{shift} = S*2^P$, where $S = sign(W); P = round(log_2(abs(W)))$
>
> We do need finetuning to mitigate the reconstruction loss and recover the accuracy. For example, both the above S and P are trainable during finetuning.
>
> If we directly reparameterize multiplication-based models with mathematical equivalent shifts **and** adds, from an algorithm perspective, there will be extreme non-uniform weight distributions that are hard to quantize (see our reply to Q1 of Reviewer qgW6); from a hardware perspective, currently efficient multiplication implementations already use shift and add units. There will be no energy or latency savings. Our current solution replaces multiplication with either shift **or** add layers, thus saving more hardware resources.
>
> ---
> **Q3: What is the weakness of applying ShiftAddNet to ViTs? Show the comparison with it.**
>
> Weakness of ShiftAddNet and qualitative comparisons (We have also included these points in Sec. 2 & 4):
>
> | ShiftAddNet | ShiftAddViT |
> |:---|:---|
> | Adopts cascaded shift layers and add layers → a doubled number of layers, parameters, and FLOPs | Keeps the original number of layers/parameters |
> | Not suitable for the MatMuls among Q/K/V in attentions as those matrices are all activations | Well supports attention in Transformers |
> | Has to train the whole model from scratch | Can inherit the pre-trained weights of ViTs |
> | No speedups on GPUs (much slower training & inference) | Provides GPU optimizations |
> | Not compatible with MoE | Compatible with MoE for switching between mixture multiplication primitives |
>
> **Quantitative comparisons.** We apply ShiftAddNet on top of PVTv2-B0 and show the comparison in ***Table 7 of the attached PDF in our global response***. We see that ShiftAddViT achieves 54%, >50x, and 59% parameters, GPU latency, and energy savings at much more stable training (ShiftAddNet suffers from loss NaN when being applied to ViTs).
>
> ---
> **Q4: Compare the quantized ViT with ShiftAddViT?**
>
> Sure, we compare our ShiftAddViT with the latest 4-bit Transformer quantization work [1] and show the results in ***Table 8 of the attached PDF in our global response***. *In terms of latency*, we see that our method achieves on average 4.7x and 2.4x GPU speedups than 4-bit quantized attentions and MLPs, respectively. *In terms of accuracy*, [1] reveals that existing 4-bit fully quantized training algorithms still have around a 1 ~ 2.5% accuracy drop on server tasks while our ShiftAddViT achieves comparable or even higher accuracy than original ViTs.
>
> [1] Training Transformers with 4-bit Integers, arXiv’23
>
> ---
> **Q5: The more complex task is better.**
>
> We follow the suggestion to extend the PVTv2-B0 (ShiftAddViT) backbone to detection (follow ViTDet [ECCV'22]) and segmentation (follow PVTv2) tasks as shown in ***Table 4 of the attached PDF in our global response***.
>
> *For the detection task*, the ShiftAdd-based backbone achieves 63.7% and 58.0% latency and energy reductions while keeping comparable mAP. *For the segmentation task*, the ShiftAdd-based backbone achieves 71.7% and 66.6% latency and energy reductions while keeping comparable mIoU.
>
> We also extend our method to NLP tasks, please refer to our reply to Reviewer FYmk's W1.

---

> > ### Comment · Reviewer_urMu · 2023-08-14
> > **Response to rebuttal**
> >
> > Thanks for your response. The rebuttal is clear and easy to understand, all my concerns have been addressed. This work is good enough to be published.

---

> > > ### Author Response · Authors · 2023-08-14
> > > **Response to Reviewer urMu**
> > >
> > > Dear Reviewer urMu,
> > >
> > > Thank you for the prompt response and for raising the rating score! We are glad all your concerns have been thoroughly addressed and will incorporate the new discussion and experiment analysis into our final revised manuscript.
> > >
> > > Best regards,
> > >
> > > Paper 9955 Authors

---

### Official Review · Reviewer_qgW6 · 2023-07-03

**Soundness:** 4 excellent
**Presentation:** 4 excellent
**Contribution:** 3 good
**Rating:** 7
**Confidence:** 5

**Summary:**

The authors propose re-parameterizing ViTs to speed up inference without a full retraining. To do this, they introduce a new operation ShiftAddViT, which works well when applied to attention. For the MLP in a transformer, the authors use a mixture of experts. This operation reduces latency and energy usage, while maintaining accuracy.

**Strengths:**

- The premise of leveraging power-of-two multiplications (a shift) is clever and a well-stated intuition (justified by Table 1 and also in L150-156). The additional energy analysis and chip-area statistics build ethos very well.
- The paper presents a very thorough set of experiments on ImageNet classification and novel view synthesis. The latency analysis is thorough and well-documented / reasoned.

**Weaknesses:**

- My only qualm is that (it appears) this relies on the existence of an effective binary quantization algorithm in L176. The results convince me that this binary quantization is acceptable, but this seems aggressive (as the authors mention later, when replacing the MLP).
- From the introduction, I initially thought this re-parameterization would be arithmetically identical to the original. After reading the methods, it appears this isn't the case? In which case, wouldn't you need some fine-tuning? If so, how costly is the fine-tuning? If I'm wrong, this may be because I don't quite understand Figure 2 -- how is this related to the original matmul?

**Questions:**

- I would've thought that the ShiftAddViT technique can really be applied to any operation (Clearly, that's not the case, as you added a modification for MLPs.) Why doesn't this work out of the box for any matrix multiplication?
- What is "ideal parallelism" in the footnote for Table 4? Are these latencies not actually measured end-to-end? If this is the case, do the baseline latency measurements also use the same assumption of ideal parallelism? How fast/slow is the MoE kernel when plugged into the network and *entire network is measured end-to-end?

**Limitations:**

See above.

The evaluation is thorough, and the method is clearly presented. Despite the questions and objections I had above, I feel this is a cogently stated idea and clearly a thoroughly-investigated problem. I do still have some questions about the method and would like to know the answers, but the idea alone is definitely worth publishing.

---

> ### Author Rebuttal · Authors · 2023-08-09
>
> We greatly appreciate your careful review and constructive suggestions. Below are our detailed responses to your concerns.
>
> **W1: The only qualm is that binary quantization seems too aggressive?**
>
> We thank the reviewer for pointing out this. We examined the sensitivity of different parts in attention blocks, where V and attention scores are much more sensitive than Q or K. Quantizing or binarizing V or attention scores yield a significant accuracy drop, e.g., a 3.6% drop when quantizing V in DeiT-T. While binarizing Q and K to convert add layers can mostly maintain the accuracy. The rationale behind this phenomenon is that Q and K mainly serve for similarity measurement while attention scores and V are the actual activations that need higher precision to maintain feature richness. That is also why we perform binarization for K in KV multiplication and Q in the Q(KV) multiplication.
>
> Our baseline Ecoformer also adopts binarization to Q and K but relies on a set of learned hash functions to map identical Q/K matrices during training. In our ShiftAddViT framework, we find that the standard layer-wise quantization is also robust for quantizing Q and K and they do not need to be identical.
>
> ---
>
> **W2: The re-parameterization would not be arithmetically identical to the original? Need finetuning? How costly is the finetuning?**
>
> Yes, you are right. We are not making our model arithmetically identical to the original ViTs. Instead, we inherit the pre-trained weight to parameterize shift or add layers. For example, we follow the below equation to reparameterize the shift weights based on the inherited weights:
>
> $W_{shift} = S*2^P$, where $S = sign(W); P = round(log_2(abs(W)))$
>
> We do need finetuning to mitigate the reconstruction loss and recover the accuracy. For example, both the above S and P are trainable during fine-tuning.
>
> If we directly reparameterize multiplication-based models with mathematical equivalent shifts **and** adds, from an algorithm perspective, there will be extreme non-uniform weight distributions (see our reply to your Q1) that are hard to quantize; from a hardware perspective, currently efficient multiplication implementations already use shift and add units. There will be no energy or latency savings. Our current solution approximates multiplications with either shifts **or** adds, thus saving more hardware resources.
>
>
> As suggested, we collect the training or finetuning wall-clock time, which is 21% ~ 25% less than training from scratch as the original Transformer model and >50x less than the previous ShiftAddNet [1] did on GPUs. For example, training the PVTv1-Tiny model from scratch takes 62 hours while our finetuning only needs 46 hours. We will add this information to the final revised manuscript to offer more insights into the training cost implications.
>
> In addition, we would like to clarify that our main goal is efficient inference and deployment, as validated by the reported real-measured inference wall-clock times on RTX 3090 GPU in the submitted manuscript. The training or finetuning cost savings are by-products as we support to reparametrize ShiftAddViTs based on the weights from already available pre-trained checkpoints of original ViTs.
>
> [1] ShiftAddNet: A Hardware-Inspired Deep Network, NeurIPS 2020
>
> ---
>
> **Q1: Why the proposed ShiftAddViT can not work out of the box for any matrix multiplication? Why add a modification to MLPs?**
>
> Good question. With W1, we are clear that our solution replaces multiplications with either shifts **or** adds instead of mathematically equivalent shifts **and** adds.
>
> The reason why ShiftAddViT cannot work out of the box is that shift- and add-based models prefer different weight distributions from multiplication-based models. For example, while repeated additions can in principle replace any multiplicative mapping, they do so in a very parameter-inefficient way. A rough example is that input 8 multiplied by weight 2 can lead to output 16, while if we only use additions to get this output, we need to add weight 8, which is much larger than the previous weights. In contrast, power-of-2 quantization in shift layers is efficient but cannot span the entire continuous space of multiplicative mappings.
>
> It is also commonly observed in the multiplication-less network community that shift- or add-based models suffer from slight accuracy drops due to less expressiveness [2] and demand additional techniques to fully recover the accuracy. This is why we propose to adopt a softer version, i.e., the mixture of experts, to mitigate the accuracy drop while preserving the benefits as much as possible to the best of our knowledge.
>
> [2] AdderNet: Do We Really Need Multiplications in Deep Learning? CVPR 2020 Oral
>
> ---
>
> **Q2: What is "ideal parallelism" in the footnote for Table 4? Do the baseline latency measurements also use the same assumption? How fast/slow is the MoE kernel when plugged into the network and the *entire* network is measured end-to-end?**
>
> The "ideal parallelism" refers to a simulated scenario to mimic parallel computing, in particular, we optimize each expert separately and measure their latency, the maximum latency among all experts will be recorded and regarded as the latency of this MoE layer, and the aggregated total of the time spent for each layer is the final reported modularized model latency.
>
> Since our baselines do not adopt the mixture of experts (MoE) so there is no need to assume "ideal parallelism". We only report the modularized latency for our models with MoE layers for readers’ reference, those latencies are not used for comparison.
>
> We want to clarify that all our comparisons are made under fair conditions, i.e., we report the end-to-end wall-clock inference time in Table 3 and use them as our final latency when compared with all baselines instead of assuming the ideal parallelism (only providing a reference number). We will make this point clear in the final revised manuscript.

---

> > ### Comment · Reviewer_qgW6 · 2023-08-13
> > **Thanks for the response**
> >
> > Thanks to the authors for their thorough response; this was very helpful, and I hadn't realized the MoE was a mixture of multiplication *or add experts. The fine-tuning cost analysis was also welcome, although I also recognize that inference is the focus here. The above addresses my questions and concerns, so I stand by my rating!
> >
> > (I think you can just submit multiple "rebuttals" per review, so the length limit per response doesn't really matter.)

---

> > > ### Author Response · Authors · 2023-08-13
> > > **Response to Reviewer qgW6**
> > >
> > > Dear Reviewer qgW6,
> > >
> > > We thank you for the timely response and for maintaining the acceptance rating! We are glad all your questions and concerns are addressed and will incorporate the new discussion and analysis into our final revised manuscript.
> > >
> > > We appreciate the given suggestions. As per this year's NeurIPS new policy, only one *"official rebuttal"* can be submitted in response to each reviewer. We will certainly follow up with *"official comments"* if other reviewers require further clarification or have additional questions.
> > >
> > > Best regards,
> > >
> > > Paper 9955 Authors

---

### Official Review · Reviewer_MmZd · 2023-07-05

**Soundness:** 3 good
**Presentation:** 3 good
**Contribution:** 2 fair
**Rating:** 5
**Confidence:** 5

**Summary:**

This paper proposes a new type of multiplication-reduced model ShiftAddViT, which use the additive kernels to reparameterize the batched GEMM in the attention block and uses the shift kernels to reparameterize other MLPs or linear layers. In this way, it can reduce energy-intensive multiplications. The authors utilize TVM to implement and optimize those kernels and achieve a speedup on GPUs, and also energy savings.

**Strengths:**

- The proposed ShiftAddViT achieves significant acceleration across different models on different tasks.
- Experiments show the real speedup on GPUs to demonstrate the effectiveness of the proposed method.


**Weaknesses:**

- Many implementation details of the proposed method are not clear enough.
- It is unfair for the comparison with DeiT. The proposed method first modified the model architecture, and then introduced the shift and add kernel to improve the model efficiency. When compared with DeiT, the architectures of those two models are not the same. The comparison should be made under the condition that the model architectures are kept the same.
- Based on Table 3 and Table 4, it appears that the “Shift” was not finally used in the classification task. And in Table 4, compared with the method that only enables “Add”, the final result (highlighted with red background) does not have any advantages in both model accuracy and latency. This seems to demonstrate that the “Shift” and “MoE” methods are not effective on classification tasks.
- Table 4 and Table 5 lack the result of method “Quant.”+”Shift”+”MoE”. It would be better if these results could be provided to see the impact on model accuracy and latency when all the proposed methods are adopted.

**Questions:**

About shift and add:
- I think the Shift method and the power-of-two quantization should be similar. But there is no scaling factor in the Shift operation (equation 3). In this way, how to convert weights in pre-trained models to the shift layers format? Is the nearest neighbor method directly adopted? Wouldn't the reconstruction error be too large without a scaling factor?
- How many bits of the P is used in the shift layers? And during finetuning, is the STE method used to solve the gradient backpropagation for s and P?
- Both two binary methods KSH[32] and Quant[26] use the scaling factors, but those scaling factors are not shown in Figure 1. During inference, where to multiply those scaling factors? Just after the add operation?
- How many bits are used for the input activations of the ShiftAddViT attention? Since only the fixed-point activations can be shifted, the input activation of shift layers needs to be quantized to an integer first. However, it seems that the article does not mention anything related to activation quantization. And there is no step related to the activation quantization in the two-stage finetuning process. In addition, if the activations are quantized, under the shift&add paradigm, where to multiply the scaling factor from activation quantization?
- How to quantize the input of the last Shift in ShiftAddViT Attention to integer values? Those input activations should also be fixed-point numbers, not floating-point.

About the MoE framework:
- In line 223, how to get the gate value $p_i$ based on the input token $x$? $p_i(x):= e^{p_i}/\Sigma_j^ne^{p_j}$ ? Is this a typo?

About the experiments:
- In Table 4, why the latency of PVT is larger than MHA? Is it because of the different batch sizes? For a fair comparison, it is better to use the same batch size here, since the model with batch_size=1 always gets worse throughput on GPUs. And it is better to show the speedup from the Linear Attention under the fair comparison.
- What’s the difference between line 5 and line 10 in Table 4? They have the same configs, but different results.
- Based on Table 4 and Table 6, “Shift” may cause an accuracy decrease, and it is necessary to introduce the “MoE” method to compensate for the model accuracy, which leads to an increase in latency. So overall, there is no advantage to using the “shift” method in terms of the model accuracy and latency. Does the “shift” method only for energy reduction?
- Experiments use the Eyeriss accelerator to get energy consumption. What are the configs of this accelerator? What data type is used for this accelerator? When counting energy, is the energy of memory access counted?

**Limitations:**

The authors discussed the limitations of the implementation of their work. It is necessary to use a customized hardware accelerator to fully leverage the benefits of “Shift” and “Add”.

---

> ### Author Rebuttal · Authors · 2023-08-09
>
> **W1: Implementation details are not clear enough**
>
> Sorry for not making it clear enough to you. We supplied more settings to Sec. 4 in the Appendix as we focused on the motivation and high-level idea in the main paper. We will clarify more and release code&models upon acceptance.
>
> **W2: For DeiT results, the architectures should be the same?**
>
> Yes, for fair comparisons, we compare to both MSA (original DeiT-T) and linear attention (LA; same architecture as ours) in ***Table 5 of the attached PDF in global response***.
>
> We see that ShiftAddViT consistently works better, reducing 43 ~ 65% and 16 ~ 43% latency/energy with comparable accuracy ($\pm$0.2%). Here LA achieves better accuracy than MSA because we adopted the Norm. (in Attn) and DWConv (in MLP) following TransNormer [EMNLP'22] & EfficientViT [ICCV'23]. The new operators also explain the increased latency despite linear complexity.
> The reason for building on LA is that Q/K is less sensitive than V/Attn (see our reply to Reviewer qgW6's W1).
>
> **W3: The shift was not finally used? No advantage of final models? Shift and MoE are not effective?**
>
> **The shift was not finally used?** We humbly clarify that the shift layer was also used in our MoE framework, i.e., Mult. expert + Shift expert, and thus was also used in the final model in Tables 3 & 4. I.e., MoE can be thought of as a soft alternative to the pure shift for parameterizing linear layers or MLPs to achieve better accuracy instead of being an orthogonal technique.
>
> **The advantage of using shift.** Replacing Mult.-based MLPs with shift layers significantly reduces latency, energy, and chip area costs. Our shift kernel offers an average speedup of 2.35x/3.07x/1.16x compared to PyTorch FakeShift, TVM FakeShift, and TVM MatMuls, respectively (Figure 3). The seemingly minor latency improvements in Tables 3 & 4 are due to full optimization of the compiled model as a whole (e.g., 6.34ms → 1ms for PVTv2-B0) on GPUs *with sufficient chip area*. Most gains are concealed by data movements and system-level schedules.
> Adopting shift layers substantially lowers energy and chip area usage (Table 1). ***Under the same chip areas***, latency savings are more pertinent, e.g., PVTv2-B0 w/ shift or MoE achieve 3.9 ~ 5.7x and 1.6 ~ 1.8x speedups, respectively, as summarized in ***Table 2 of the attached PDF in our global response***.
>
> **MoE to compensate for accuracy drop.** MoE is a soft alternative to adopt both Mult. and shifts to achieve a better accuracy-efficiency trade-off, i.e., on average +1.36% accuracy gain with 46% ~ 53% shift benefits preserved.
>
> **W4: Lack the results of “Quant. + Shift + MoE”.**
>
> We conduct the requested experiment for PVTv2-B1:
> 1. Quant.: 78.70%
> 2. Quant. + Shift (Attn & MLPs): 77.55%
> 3. Quant. + Shift (Attn) + MoE (MLPs): 78.23%.
>
> This is also consistent with our previously reported results in lines 5-7 of Table 6 when adopting KSH instead of Quant: "KSH + Shift (Attn) + MoE (MLPs)": 78.20%.
>
> We humbly clarify that MoE is an alternative to Shift and they cannot be applied for the same layer. Let us know if we misunderstand your question.
>
> **Q1: Implementation details about the shift & add?**
>
> **Scaling factor for shifts? How to convert pre-trained weights?** We reparameterize shifts following DeepShift-PS [CVPRW'21] and do not use a scaling factor.
>
> $W_{shift} = S*2^P$, where $S = sign(W); P = round(log_2(abs(W)))$
>
> As both S & P are trainable during the finetuning, the reconstruction loss will be reduced.
>
> **Bit allocation? STE used?** We adopt 4 bits for P and yes, STE is used following DeepShift.
>
> **Where to multiply scaling factors for KSH and Quant.?** For KSH, there is no scaling factor needed as a set of hash functions is applied to convert Q/K to binary codes. For Quant., we leverage layer-wise Quant. for both Q & K, the scaling factor can be multiplied after add ops. It can be efficiently implemented following Sec. 2.2 of [1].
>
> **Input activation Quant. for shifts? Last Shift?** The input activation of **all** shift layers is rounded to 16 bits of fixed-point precision format following DeepShift. We use layer-wise Quant. so the scaling factor can be multiplied after shift ops.
>
> [1] Quant. & Training of NNs for Integer-Arithmetic-Only Inference, CVPR'18
>
> **Q2: How to get gate value in MoE?**
>
> Sorry for the confusion as we merge both gate and softmax into one equation. The gate itself is a linear layer and p = G(x) is the output gate value of dimension 2 (number of experts). The softmax is for normalizing p, and for efficiency, we use argmax to select one expert.
>
> **Q3: Experiment details?**
>
> **PVT slower than MSA? Different batch sizes (BS)?** We ensured BS=1 for all latency measurements and also found this counterintuitive phenomenon.
>
> The reasons are two folds: (1) linear attention introduces extra ops, e.g., normalization, DWConv, or spatial reduction block that cost more time under small BS; and (2) the linear complexity is w.r.t. the number of tokens while the input resolution of 224 is not sufficiently large → limited benefits.
>
> To validate both points, we measure the latency of PVTv2-B0 with various BS and input resolutions as shown in ***Table 3 of the attached PDF in our global response***: Linear attention’s benefits show up under larger BS or input resolutions.
>
> **Difference between lines 5 & 10 in Table 4?** For line 5, we use KSH where Q & K are identical following Ecoformer; For line 10, Q & K are independent as we directly quantize both.
>
> **Shift advantages? Only energy savings?** No, using shifts has comprehensive benefits. Please refer to our reply to your W3 for the analysis.
>
> **Configs of Eyeriss? What data types? Memory access?** The configs like bit allocations are matched with our algorithm (e.g., INT32 for adds; INT16 for shifts). Data type and unit energy are reported in Table 1. We do count the memory access costs. More details can be found in DNN-Chip-Predictor [ICASSP'20] of which the contribution is the simulator of Eyeriss.

---

> > ### Comment · Reviewer_MmZd · 2023-08-14
> >
> > Thanks for the feedback. My concerns are mostly well addressed, so I raise my rating. I hope the authors could add these detailed illustrations about model quantization to the revised paper, to make it clearer and stronger.

---

> > > ### Author Response · Authors · 2023-08-14
> > > **Response to Reviewer MmZd**
> > >
> > > Dear Reviewer MmZd,
> > >
> > > Thank you for the prompt response and for raising the rating score! We are glad your questions and concerns are well addressed and will certainly include the detailed quantization illustrations in our final revised manuscript.
> > >
> > > Best regards,
> > >
> > > Paper 9955 Authors

---

### Official Review · Reviewer_hFZt · 2023-07-07

**Soundness:** 3 good
**Presentation:** 4 excellent
**Contribution:** 3 good
**Rating:** 6
**Confidence:** 3

**Summary:**

This paper introduces a novel reparameterization method for efficient Vision Transformers (ViT). The method replaces the heavy multiplication operations in ViT with a combination of shift and add operations. By mapping queries and keys to binary codes in Hamming space and reparameterizing multi-layer perceptrons (MLPs) or linear layers, the multiplication operations in the model are effectively reduced. Experimental results demonstrate that ShiftAddViT achieves efficient performance on various 2D/3D Transformer visual tasks while achieving latency reduction and energy savings.

**Strengths:**

This paper presents a clear motivation for each proposed compound. For example, the reparameterization of multiplications is inspired by hardware design practices and the concept of multiple experts (MOE) for dynamically routing tokens into multiplication or shift groups. The three questions posed by the authors clearly indicate the specific problems they are addressing. The writing and demonstration are clear and concise. In terms of results, this work delivers some impressive findings in regard to both efficiency and accuracy. Extensive ablation experiments are conducted to validate the proposed approach further.

**Weaknesses:**

1. A primary concern is the limited validation of the method proposed by the authors, which is solely performed on small-scale models, with the largest model encompassing 30M and 4G FLOPs. I am particularly interested in the performance of larger-scale models. Naturally, I acknowledge the constraints imposed by computational resources. However, considering the hardware specifications disclosed by the authors, it is apparent that training models of ViT-Base size or similar are feasible.

2. Secondly, my concern lies in the stability of the training process and the associated wall-clock time. Obtaining information regarding the overall training cost and wall-clock time from the authors would offer valuable insights into the algorithm's feasibility and cost implications.

**Questions:**

Overall, I am satisfied with the results and motivation.   I am particularly concerned about whether this method can improve inference performance for larger models. The bottleneck for small models is not particularly high, to begin with, and compared to larger models, the performance of small models will undoubtedly be lower. Therefore, if the authors' method can enable larger models to achieve a significant increase in inference speed, it would greatly enhance the significance of the approach.

**Limitations:**

The author discussed the limitation in their paper.

---

> ### Author Rebuttal · Authors · 2023-08-08
>
> We greatly appreciate your careful review and constructive suggestions. Below are our detailed responses to your concerns.
>
> **W1: Most experiments are small-scale models (largest: 30M parameter and 4G FLOPs), how about the performance of larger-scale models like ViT-Base size or similar?**
>
> Thank you for your suggestion. As you advised, we further examine our proposed method on large-scale models, including PVTv2-B5 (Params: 82M; FLOPs: 11.8G) and DeiT-Base (Params: 86M; FLOPs: 17.6G) on ImageNet. The results are shown in the table below as well as in ***Table 6 of the attached PDF in our global response***. We can see that the proposed ShiftAddViT consistently performs better in terms of accuracy-efficiency tradeoffs, achieving 18.4% ~ 65.7% and 28.9% ~ 70.3% latency/energy reduction with comparable accuracy ($\pm$0.5%).
>
> | Models | Methods | Accuracy (%) | Latency (ms) | Energy (mJ) |
> |---|---|:---:|:---:|:---:|
> | DeiT-Base | Linear Attention | 83.1 | 8.43 | 625.74 |
> | DeiT-Base | **ShiftAddViT** | 82.9 | 6.88 | 185.80 |
> | PVTv2-B5 | Linear SRA | 83.8 | 39.80 | 482.94 |
> | PVTv2-B5 | **ShiftAddViT** | 83.3 | 13.66 | 343.37 |
>
> As we mainly target mobile and small-to-medium model scale scenarios previously, this new set of large model experiments further validates the scalability and potential of our ShiftAddViT on large model settings. We will incorporate the new results into the revised manuscript.
>
> ---
>
> **W2: Stability of the training process and the associated wall-clock time?**
>
> The process of training or finetuning after reparameterizing Transformer models using shifts and adds demonstrates stability. We conducted experiments involving two-step finetuning. In the initial step, we transformed Multi-Head Self Attention (MSA) into linear attention and reparameterized all Matrix Multiplications (MatMuls) with additive layers. This was followed by finetuning to restore accuracy. In the subsequent step, we reparameterized MLPs or linear layers using shift or MoE layers, again finetuning for accuracy recovery.
>
> As suggested, we collect the training or finetuning wall-clock time, which is 21% ~ 25% less than training from scratch as the original Transformer model and >50x less than the previous ShiftAddNet [1] did on GPUs. For example, training the PVTv1-Tiny model from scratch takes 62 hours while our finetuning only needs 46 hours. We will add this information to the final revised manuscript to offer more insights into the training cost implications.
>
> In addition, we would like to clarify that our main goal is efficient inference and deployment, as validated by the reported real-measured inference wall-clock times on RTX 3090 GPU in the submitted manuscript. The training or finetuning cost savings are by-products as we support to reparametrize ShiftAddViTs based on the weights from already available pre-trained checkpoints of original ViTs.
>
> [1] ShiftAddNet: A Hardware-Inspired Deep Network, NeurIPS 2020

---

> > ### Comment · Reviewer_hFZt · 2023-08-15
> > **Thanks for the rebuttal**
> >
> > Thanks for the larger models' experiments. Although we can see the performance has a potential drop, the improvement in inference time is notable. My concerns have been mainly addressed. I have raised my score.

---

> > > ### Author Response · Authors · 2023-08-15
> > > **Response to Reviewer hFZt**
> > >
> > > Dear Reviewer hFZt,
> > >
> > > Thank you for the prompt response and for raising the rating score! We are glad your concerns are addressed and will include the new results and corresponding analysis in our final revised manuscript.
> > >
> > > Best regards,
> > >
> > > Paper 9955 Authors

---

### Official Review · Reviewer_FYmk · 2023-07-07

**Soundness:** 4 excellent
**Presentation:** 4 excellent
**Contribution:** 3 good
**Rating:** 6
**Confidence:** 4

**Summary:**

This work proposes ShiftAddViT, which is an efficient ViT reparameterization with a mixture of complementary multiplication primitives, such as bitwise shifts and adds.
The alternative parameterization (quantization) is carefully examined to allocate to different components (MHSA, MLP) in ViT.
In addition, the authors propose a mixture of experts (MoE) framework to classify input tokens and assign different primitives to best preserve accuracy.
The MoE framework is guided by latency-aware load-balancing loss.
Multiple empirical results are demonstrated including 2D ViT in image classification task, as well as GNT for NVS task.

**Strengths:**

1. The paper is clearly written and easy to follow.
2. Though shift and add reparameterization is not completely new (binary, ternary or power-of-2 quantization), this is a very pioneer work to systematically investigate layer sensitivity and accuracy impact, allocation strategy, and hardware implementations. The hardware benchmarks with considerable latency and energy savings are impressive.
3. The experimental analysis is strong. In addition to conventional classification task, the authors also provide results on NVS task.

**Weaknesses:**

1. Since this work proposes a collection of analysis and optimizations on MHSA and MLP, reducing the computation complexity and energy consumption, I wonder if this work is portable to NLP tasks, especially for LLMs with more tokens?
2. I cannot fully follow the token dispatching method mentioned in Section 4.2 and Figure 6. The authors did not discuss the methodology for input allocation and parallel computing issues in detail (line#227). Intuitively, if the important and sensitive tokens (yellow) in Figure 6 are fed into powerful experts (MULT.), while the rest are into SHIFT experts, are they still visible to each other (i.e. global receptive field)?

**Questions:**

Please refer to concerns raised in weaknesses.

---

> ### Author Rebuttal · Authors · 2023-08-08
>
> We greatly appreciate your positive comments and constructive suggestions. Below are our detailed responses to your concerns.
>
> **W1: I wonder if this work is portable to NLP tasks, especially for LLMs with more tokens?**
>
> We follow your suggestion to test our proposed optimization of MHSA and MLPs to NLP tasks. In particular, we apply our methods to Transformer models and Long Range Arena (LRA) benchmarks consisting of sequences ranging from 1K to 16K tokens [1]. The results are shown in the table below as well as in ***Table 1 of the attached PDF in our global response***.
>
> | Models | Listops (2K) | Retrieval (4K) | Text (4K) | Image (1K) | Average Accuracy | Latency (ms) | Energy (mJ) |
> |---|---|:---:|:---:|:---:|:---:|:---:|:---:|
> | Transformer | 37.10 | 79.35 | 65.02 | 38.20 | 54.92 | 84.54 | 139.83 |
> | Reformer | 19.05 | 78.64 | 64.88 | 43.29 | 51.47 | 11.19 | 19.04 |
> | Linformer | 37.25 | 79.37 | 55.91 | 37.48 | 52.59 | 12.13 | 19.68 |
> | Performer | 18.80 | 78.62 | 63.81 | 37.07 | 49.58 | 11.93 | 18.74 |
> | **ShiftAdd-Transformer** | 37.15 | 82.02 | 66.69 | 35.62 | **55.37** | **7.38** | **8.53** |
>
> The results consistently show the superior performance of our proposed ShiftAdd Transformers in terms of both model accuracy (+0.45% ~ +5.79%) and efficiency (1.5x ~ 11.5x latency reductions and 2.2x ~ 16.4x energy reductions on an Eyeriss-like accelerator) as compared to original Transformer and other linear attention baselines, which means that our shift and add reparameterization and load-balanced MoE ideas are generally applicable to Transformer models and are agnostic to domains and tasks. We thank the reviewer for this comment as this set of experiments also enlarges the impact of our work.
>
> [1] Long Range Arena: A Benchmark for Efficient Transformers, ICLR 2021
>
> ---
>
> **W2: How to understand the dispatching method in Section 4.2 and Figure 6? Discuss the methodology for input allocation and parallel computing issues in detail (line#227), are input tokens of multiplication experts and shift experts still visible to each other (i.e., global receptive field)?**
>
> Thanks for pointing out this question, we will clarify each point one by one.
>
> ***How to understand the dispatching.*** As for the dispatching method, our hypothesis is that important and sensitive tokens are expected to be handled by powerful multiplication experts while the rest are into cheaper shift experts as you also pointed out. The trainable router within the MoE layers automatically learns this dispatch assignment as the gradients decay and the loss converges to minima in loss landscapes. This learning process is guided by our proposed latency-aware load-balancing training loss function (integrated with classification loss) in Section 4.2. Figure 6 visualizes the actual learned dispatching pattern in our practically trained model to verify our hypothesis.
>
> ***Elaborate input allocation and parallel computing.*** Sorry for not fully expanding it due to the limited space. We elaborate on these two points below.
>
> 1. *For input allocation*, it is determined dynamically during runtime, we know the allocation only when the model is executed and we receive the router outputs. Therefore, the shape of expert input and corresponding indexes are dynamically changed. This can be handled by PyTorch with dynamic graphs while TVM expects static input shape. That is why we leverage the compiler support for dynamism as proposed in Nimble [2] on top of the Apache TVM to handle the dynamic input allocation.
>
> 2. *For parallel computing*, it means that different experts are run in parallel, this can be supported by several customized distributed training frameworks integrated with PyTorch, e.g., FasterMoE [3], and DeepSpeed [4]. In contrast, it remains nontrivial to support this in the TVM community. One option to simulate is to perform modularized optimization to mimic parallel computing, in particular, we optimize each expert separately and measure their latency, the maximum latency among all experts will be recorded and regarded as the latency of this MoE layer, and the aggregated total of the time spent for each layer is the final reported modularized model latency. To avoid any potential confusion between real-measured wall-clock time, i.e., no parallelism assumed, and simulated modularized latency, i.e., ideal parallelism assumed, we reported both for models containing MoE layers as shown in Tables 4 and 6 to offer more insights into the algorithm’s feasibility and cost implications.
>
> ***Maintain global receptive field.*** Good point, the input tokens of different experts are still visible to each other because we only split tokens for MLP layers while keeping the attention mechanism untouched, so that all tokens still attend to each other in attention layers to ensure the global receptive field. In addition, for PVTv2 models, there are depthwise convolutions in the middle of two MLP layers to exchange information between two experts’ outputs.
>
> We appreciate the reviewer for raising the potential confusion readers may have and will clarify all these points in the final revised manuscript.
>
> [2] Nimble: Efficiently Compiling Dynamic Neural Networks for Model Inference, MLSys 2021
>
> [3] FasterMoE: Modeling and Optimizing Training of Large-Scale Dynamic Pre-Trained Models, ACM SIGPLAN Symposium on Principles and Practice of Parallel Programming 2022
>
> [4] DeepSpeed-MoE: Advancing Mixture-of-Experts Inference and Training to Power Next-Generation AI Scale, ICML 2022

---

> > ### Comment · Reviewer_FYmk · 2023-08-16
> >
> > Thanks for the rebuttal. My concerns are well addressed, thus I keep my rating.

---

> > > ### Author Response · Authors · 2023-08-16
> > > **Response to Reviewer FYmk**
> > >
> > > Dear Reviewer FYmk,
> > >
> > > We thank you for the timely response and for keeping the acceptance rating! We are glad all your concerns are addressed and will incorporate the new experimental results and analysis into our final revised manuscript.
> > >
> > > Best regards,
> > >
> > > Paper 9955 Authors

---

### Author Rebuttal · Authors · 2023-08-09

**Dear ACs and Reviewers,**

First of all, we deeply appreciate the time and effort spent by you in providing the reviews, and truly value your effort, especially considering the substantial scale of a conference like NeurIPS.

We are immensely grateful for the positive feedback our paper has received. The accolades, including remarks about its excellent soundness and presentation, pioneering systematic investigation, impressive achievements in latency and energy savings, and the clarity and conciseness of both the writing and demonstrations, alongside the extensive and thorough experiments, are all deeply gratifying. It's particularly encouraging that these aspects have garnered unanimous appreciation from the reviewers.

Despite the commendations, we have also received inquiries from reviewers requesting additional experiments and further clarification. We have supplied the requested experiments and provided detailed clarifications to raised questions as summarized below.

---
**To summarize, the following experiments have been supplied:**

- *Extend to NLP tasks and more token scenarios*
  - We have included the results in our rebuttal response to *Reviewer FYmk's W1*.
  - Also refer to *Table 1* of the attached PDF for the organized result table.
- *Extend to larger models like ViT-Base or similar*
  - We have included the results in our rebuttal response to *Reviewer hFZt’s W1*.
  - Also refer to *Table 6* of the attached PDF for the organized result table.
- *DeiT comparison under the same architecture*
  - We did not provide a result table in our response to *Reviewer MmZd's W2* due to length limitations.
  - Please refer to *Table 5* of the attached PDF for the organized result table.
- *Ablation studies of adopting shift or not*
  - We did not provide a result table in our response to *Reviewer MmZd's W3* due to length limitations.
  - Please refer to *Table 2* of the attached PDF for the organized result table.
- *Ablation studies of varying batch sizes and input resolutions*
  - We did not provide a result table in our response to *Reviewer MmZd's Q3* due to length limitations.
  - Please refer to *Table 3* of the attached PDF for the organized result table.
- *Add baseline: compare to ShiftAddNet*
  - We did not provide a result table in our response to *Reviewer urMu’s Q3* due to length limitations.
  - Please refer to *Table 7* of the attached PDF for the organized result table.
- *Add baseline: compare to quantized ViTs*
  - We did not provide a result table in our response to *Reviewer urMu’s Q4* due to length limitations.
  - Please refer to *Table 8* of the attached PDF for the organized result table.
- *Extend to more complex tasks, such as detection and segmentation*
  - We did not provide a result table in our response to *Reviewer urMu’s Q5* due to length limitations.
  - Please refer to *Table 4* of the attached PDF for the organized result table.

---
**To summarize, the following questions have been clarified:**

- *MoE dispatching and the methodology for input allocation and parallel computing*
  - We clarify this in our response to *Reviewer FYmk’s W2 and qgW6’s Q2*.
- *Training stability and wall-clock time*
  - We clarify this in our response to *Reviewer hFZt’s W2 and qgW6’s W2*.
- *Implementation details*
  - Scaling factors, bit allocation, and gate mechanism in ShiftAddViT
    - We clarify this in our response to *Reviewer MmZd’s Q1 and Q2*.
  - Difference between line 5 and line 10 in Table 4, i.e., KSH vs. Quant
    - We clarify this in our response to *Reviewer MmZd’s Q3*.
  - Configs of Eyeriss accelerator
    - We clarify this in our response to *Reviewer MmZd’s Q3*.
  - Lack the results of “Quant. + Shift + MoE”.
    - We clarify this in our response to *Reviewer MmZd’s W4*.
- *The advantages and rationale of adopting shifts and MoE; The contributions of multiplication less and MoE are independent?*
  - We clarify this in our response to *Reviewer MmZd’s W3/Q3 and urMu’s W1/Q1*.
- *Why PVT or Linear attention is slower than MSA*
  - We clarify this in our response to *Reviewer MmZd’s Q3*.
- *Binary quantization seems too aggressive*
  - We clarify this in our response to *Reviewer qgW6’s W1*.
- *The re-parameterization would not be arithmetically identical to the original?*
  - We clarify this in our response to *Reviewer qgW6’s W2/Q1 and urMu’s Q2*.
- *Suitable for algorithm-based conferences like NeurIPS/CVPR/ICML?*
  - We clarify this in our response to *Reviewer urMu’s W1*.

---

Regarding Reviewer MmZd's questions about implementation details, we've clarified concisely within the rebuttal's length limitations. We are open to providing further details if any points still seem unclear. To enhance reproducibility, we will release both the codebase and pre-trained models, enabling others to replicate our results effectively.

We would be much appreciated if you could check our rebuttal response and expect the new experiments and clarifications can solve your concerns. We are always willing to be involved in the discussion with you, please let us know if our responses do not resolve your concerns so that we can further clarify, thanks!

Best regards,

Paper 9955 Authors

---

### Decision · Program_Chairs · 2023-09-21

**Decision:**

Accept (poster)

**Comment:**

All reviewers support the acceptance of this paper. The initial concerns have been addressed during the rebuttal.